

# Regional PM$_{2.5}$ pollution confined by atmospheric internal boundaries in the North China Plain: 2. boundary layer structures and numerical simulation

Xipeng Jin[1], Xuhui Cai[1]*, Mingyuan Yu[2], Yu Song[1], Xuesong Wang[1], Hongsheng Zhang[3], Tong Zhu[1]

[1]College of Environmental Sciences and Engineering, State Key Lab of Environmental Simulation and Pollution Control, Peking University, Beijing 100871, China

[2]School of Applied Meteorology, Nanjing University of Information Science and Technology, Nanjing 210044, China

[3]Department of Atmospheric and Oceanic Sciences, School of Physics, Peking University, Beijing 100871, China

*Correspondence to*: Xuhui Cai (E-mail: xhcai@pku.edu.cn)

**Abstract.** This study reveals and summarizes mesoscale planetary boundary layer (PBL) structures for
various pollution patterns in the North China Plain. Three pollution categories have been classified, in
terms of the influence of the atmospheric internal boundary (AIB) that significantly determines the
distribution and concentration of PM$_{2.5}$. The Weather Research and Forecast model is used to simulate
the PBL structure in this region, and its performance is firstly evaluated using surface observations and
intensive soundings data. Observed AIBs and PBL evolution are reasonably reproduced. Simulation
results for three pollution categories illustrate respective PBL structures, as well the relationship with the
mesoscale AIBs. The first category corresponds to the severest pollution and occurs most frequently
(~41 %). The PBL structure is laterally confined by a warm front as a sharp AIB and vertically suppressed
by a dome-like elevated temperature inversion, which constitutes a stable and enclosed circumstance,
most favorable to pollution formation. The second category is characterized by wind shear line/zone as
AIB, with dynamic convergence in the PBL as the dominant cause for PM$_{2.5}$ accumulation. Three shear
modes consist of this category, two of which are related to pressure troughs with the convergence layer
of the order of the PBL depth. Another shear mode presents a much thicker convergence layer with a
depth of about 3000 m, under the saddle-shaped pressure field. This category corresponds to lighter air
pollution, with a frequency of 29 %. The PBL of the third category is laterally delineated by a cold-air
damming AIB at the foot of the mountains on the windward side. It manifests as a low-temperature and
weak-wind air mass accompanied by an elevated inversion and a convergent flow with a thickness as
high as mountains. This PBL structure maintains through day and night within the AIB confined zone,
while the ordinary diurnal variation of the PBL occurs outside this zone. 14 % of pollution episodes



belong to this category. There remain about 16 % pollution episodes undefined by the AIB influence.
They may need to be analyzed separately in the future.
**Keywords**: Boundary layer structure; atmospheric internal boundaries; $PM_{2.5}$; modeling
**1 Introduction**
The planetary boundary layer (PBL) is the lowest section of the atmosphere that responds directly
to the heat and friction from the Earth's surface (Stull, 1988; Garratt, 1992). Most air pollutants are
intensively emitted or chemically produced within this layer, and their horizontal transport and vertical
mixing are affected by the dynamic flow and thermal stability of the PBL (Tennekes, 1974). Therefore,
the PBL structure is a crucial role in the evolution, magnitude and distribution of air pollution.
The PBL structure has been recognized to be strongly dependent on three categories of factors: (i)
the single-column vertical property forced by the local surface's energy balance; (ii) the lateral-section
horizontal variation regulated by the mesoscale meteorological process and (iii) the three-dimensional
spatial evolution controlled by the large-scale synoptic system (Boutle et al., 2010). The local vertical
PBL structure and its impact on air pollution have been widely discussed from different aspects including
turbulent mixing (Emeis and Schafer, 2006; Ren et al., 2019), dynamic effect (Dupont et al., 2016),
entrainment (Li et al., 2018; Jin et al., 2020), and radiative feedback with aerosol (Petaja, 2016). In these
studies, the PBL height at a certain site has been the most commonly used indicator to analyze the
correlation with pollutant concentration, whether from the time scale of the diurnal cycle, daily variation,
or longer period (Bianco et al., 2011; Liu et al., 2019; Miao and Liu, 2019). Moreover, some studies
investigate the PBL spatial structure under the large-scale force of weather systems (Prezerakos, 1998;
Boutle et al., 2010; Mayfield and Fochesatto, 2013). Sinclair et al. (2010) report the three-dimensional
PBL structure developed beneath an idealized mid-latitude weather system, which is characterized by a
deep convective PBL in the eastern flanks of the anticyclone and a shallow shear-driven PBL in the
cyclone's warm sector. The effect of the monsoon trough on the PBL has also been indicated, showing
relatively low PBL capped by a stable layer in the western end of the trough line, while a well-defined
deep moist layer with active thermal instability in the eastern end (Rajkumar et al., 1994; Narasimha,
1997; Potty et al., 2001). In recent years, synoptic classification has been used to explore the role of
different weather circulations on PBL structure and to further analyze air pollution (Peng et al., 2016;
Xiao et al., 2020). The movement of the synoptic systems makes the shallow and deep boundary layers
develop alternately in a certain area, regulating the periodic evolution of large-scale air pollution.
As the intermediate, mesoscale systems interact with PBL in more direct and complex ways, since
they occur in the low-level troposphere with vertical extension comparable with the PBL depth and
horizontal scale closing to the regional range. Discontinuity of meteorological properties inside and





outside these systems presents as atmospheric internal boundary (AIB) in the lateral direction, leading to
the abrupt change of the PBL spatial structure, which is of particular importance to the formation and
maintenance of regional pollution. The effects of mesoscale sea-land and mountain-valley circulations
on the PBL have been clarified, i.e., the thermal internal boundary layer in the coastal area and the
depressed PBL close to a mountain base (Garratt, 1990; Lu and Turco, 1995; Talbot et al., 2007; De
wekker, 2008; Miao et al., 2015). Some studies discuss the PBL structure under the rule of other types
of mesoscale/sub-synoptic scale systems, such as the persistent cold-air pools in the Salt Lake valley
(Lareau et al., 2013), foehn winds in the Eastern Alps (Seibert, 1990; Baumann et al., 2001), and leeside
troughs and cold-air damming around the Appalachian mountains (Seaman and Michelson, 2000; Bell
and Bosart, 1988), as well as the frequent cold and warm fronts in Europe (Berger and Friehe, 1995;
Sinclair, 2013). However, there needs more understanding of their impact on the evolution of air pollution.

The North China Plain (NCP) is one of the most polluted areas in the world, to which extensive

investigation has been conducted. Nevertheless, the knowledge about the PBL spatial structures under
the impact of the mesoscale systems and the role AIBs play in this region is still insufficient. Based on
the surface observations, a thorough survey of the $PM_{2.5}$ pollution categories under the control of the
AIBs is carried out in a companion paper (Jin et al., 2022 submitted). It is found that the pollution
formation-maintenance process in the NCP can be classified into three categories, i.e., the frontal
category, wind shear category and topographic obstruction category respectively. The first category
represents about 41 % of all episodes during the autumn and winter of the investigated 7 years (2014–
2020). An isolated cold area is bounded by a warm front, which plays as the AIB. The second category
is determined by dynamic wind shears. Three modes of AIBs are characterized by west-southwest wind
shear, southeast-east wind shear and south-north wind shear respectively. The third category is closely
related to the cold-air damming effect with the AIB formed between the prevailing airflow and the
blocked air toward the terrain. Although the results in Jin et al. (2022 submitted) clearly demonstrate the
relationship of the surface AIBs to the pollution episodes and their spatial patterns, the three-dimensional
structures of these mesoscale AIBs and their interplay with the PBL are not yet clarified, those are
believed to be of critical significance to the regional pollution. The present study tries to fulfill this
knowledge gap.

The mesoscale meteorological models, such as the Weather Research and Forecast (WRF) with the

high spatial and temporal resolution, are plausible tools to capture the mesoscale systems and display
detailed spatial structures in the lower atmosphere, including the AIBs and the PBL (Jimenez et al., 2016,
Pielke and Uliasz, 1998; Seaman, 2000; Hanna and Yang, 2001; McNider and Pour-Biazar, 2020). The
present study aims to reveal the thermal and dynamic structures of the PBL and their evolution associate
with different types of AIBs in the condition of pollution episodes, by using the WRF model simulations.
For this purpose, the model performance is at first evaluated with detailed sounding data from the





intensive experiment, to ensure the model's ability in reproducing the meteorological fields and their
three-dimension structures in the concerned region. The article is organized as follows. The following
section describes the PBL sounding observations as well as the WRF simulation. Section 3 provides an
overview of representative pollution cases and the evaluation of the model performance. Furthermore,
the PBL spatial structure under each pollution category is analyzed. Finally, the conclusions are presented
and the feasibility and limitation of mesoscale models are discussed in Sect. 4.
**2 Data and methods**
**2.1 Observations and data analysis**
**Intensive GPS (Global Positioning System) sounding data**: Two periods of field experiments
were carried out to evaluate the meteorological model and explore wintertime PBL structure in the NCP:
at Cangzhou (38°13′ N, 117°48′ E, Fig. 1a) from January 8 to 28, 2016 and at Dezhou (37°16′ N, 116°43′
E, Fig. 1a) from December 25, 2017, to January 24. GPS radiosonde (Beijing Changzhi Sci & Tech Co.
Ltd., China) was used to obtain profiles of wind speed, wind direction, temperature and relative humidity
with a vertical resolution of approximately 1 s (3~5 m). Eight soundings were taken on each day, at 0200,
0500, 0800, 1100, 1400, 1700, 2000 and 2300 LT (i.e., UTC + 8). The reliability of the GPS sounding
data has been systematically evaluated by Li et al. (2020) and Jin et al. (2020, 2021).
**Routine radiosonde sounding data:** Routine sounding data from the meteorological station of
Beijing (39°56′ N, 116°17′ E, Fig. 1a) was collected in the absence of intensive PBL observation. The
data were obtained from Wyoming University, USA (http://weather.uwyo.edu.html), and the original
observation data came from the China Meteorological Administration. The routine soundings were taken
2 times a day, at 0800 and 2000 LT.
**PBL height and vertical profiles:** During the two periods of intensive field experiments, 160 and
240 datasets were collected at each site, including vertical profiles of temperature, relative humidity,
wind speed, and wind direction. We carried out quality control on the original sounding data and
eliminated outliers and then calculated the profiles of potential temperature. All the profiles were
smoothed by the three-point moving average method and were interpolated to obtain a vertical resolution
of 10 m. The PBL height was derived via the potential temperature profile method and the detailed
calculation followed the mathematical method established by Liu and Liang (2010). Sounding data were
used to evaluate model performance and to analyze the three-dimensional thermal and dynamic spatial
structure of the PBL.
In addition to the PBL sounding data, the routine meteorological observation and air quality
monitoring data were used to obtain the surface meteorological field and pollutant concentration field.
The spatial distributions of sea level pressure, 10 m wind vector, potential temperature and the



corresponding PM$_{2.5}$ concentration were obtained by data interpolation or diagnostic model, details of
the methods referred to Jin et al. (2021).

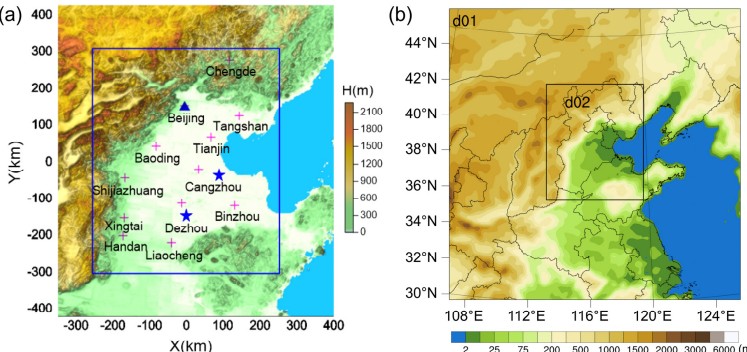


Figure 1. Geographical map of the (a) observation area and (b) WRF model domain. Intensive GPS
soundings at Dezhou and Cangzhou (pentagram), routine radiosonde sounding at Beijing (triangle) and
air quality stations (plus) are indicated in (a). The rectangle in (a) is the same as the model inner domain
d02 in (b).

**2.2 Model simulations**

The WRF model was used to investigate the vertical and horizontal structures of the PBL. Two
nested domains (Fig. 1b) were employed with horizontal grid resolutions of 15 and 5 km. Each domain
had 37 vertical layers extending from the surface to 100 hPa, with 25 layers within 2 km to resolve the
PBL structure. The meteorological initial and boundary conditions were set using the United States
National Center for Environmental Prediction Final Analysis (NCEP-FNL) dataset. The physics
parameterization schemes applied in this study were the same as Jin et al. (2021).

**2.3 Representative cases**

As mentioned above, three categories, six types of PM$_{2.5}$ pollution episodes associated with
mesoscale AIBs have been identified in the NCP (Jin et al. 2022 submitted). The present study tries to
reveal the PBL structures and their evolution for these pollution and AIB types. Typical cases
representative of the respective types were selected for this purpose. For the first category, two frontal
types shared a similar PBL structure and have been investigated previously (Jin et al., 2021), which would
be recapitulated in the following section. In the second category, the southeast-east wind shear type had
a very low occurrence frequency (4 %) and showed similar characteristics to the west-southwest wind
shear type. Therefore, the two main types of wind shear category and topographic obstruction category
were investigated in this paper. Three typical cases/episodes were selected to respectively represent the
corresponding pollution types, i.e., Case-1 for west-southwest wind shear type: during January 18–21,





2018; Case-2 for south-north wind shear type: during January 7–11, 2016; and Case-3 for topographic
obstruction type: during October 7–12, 2014. The temporal and spatial evolution of their $PM_{2.5}$
concentrations and the corresponding surface meteorological conditions would be analyzed based on
routine observations, and their PBL spatial structures would be revealed by the WRF model simulations.
**3 Results**
**3.1 Basic features of the cases**
The surface observations for these three cases are presented firstly. According to the temporal
evolution of $PM_{2.5}$ concentration at different stations in the NCP (Fig. 2), all of these three pollution
episodes went through the stages of formation, maintenance and diffusion. As shown in Fig. 2a, Case-1
was characterized by two main peaks in the formation-maintenance stage (January 18–20, 2018), with
the latter being higher than the former (500 μg m$^{-3}$ at Canzhou vs 300 μg m$^{-3}$ at Handan). From noon on
January 20, 2018, pollution in Tianjin-Cangzhou-Shijiazhuang diffused successively and all sites reached
a clean level on the afternoon of January 21, 2018. For Case-2, the pollution formed in the first two days,
maintained over the next day and was cleaned on the night of January 10, 2016 (Fig. 2b). The southern
sites such as Liaocheng and Dezhou were the most polluted (reaching 450 μg m$^{-3}$) and the northern cities
such as Beijing and Chengde were the least polluted (less than 150 μg m$^{-3}$). Pollution in Case-3
experienced the formation process on October 7–8, 2014, maintained for the successive three days, and
ended on October 12, 2014 (Fig. 2c). During this period, the piedmont sites (Baoding, Beijing and
Shijiazhuang) kept always a high concentration regardless of day and night (about 400 μg m$^{-3}$), while
the southeast sites (Binzhou, Dezhou and Cangzhou) had lighter pollution and obvious diurnal cycle
(lower than 250 μg m$^{-3}$).
The spatial patterns of $PM_{2.5}$ pollution, from the formation (Fig. 3i), maintenance (Fig. 3ii-iv), to
diffusion stage (Fig. 3v), are illustrated for each case. In the formation stage, the polluted air mass of
Case-1 and Case-3 built up along the mountains from the southwest of the NCP (Fig. 3a-i, c-i) while it
was located more south in Case-2 (Fig. 3b-i). During the pollution maintenance process, Case-1 was
accompanied by widespread $PM_{2.5}$ flooding the NCP, during which the heaviest pollution center has been
transferred eastward (Fig. 3a, ii-iv); in Case-2, a polluted air mass has been advancing northward with a
clear edge, but it did not reach the northern mountainous area (Fig. 3b, ii-iv); the spatial distribution of
$PM_{2.5}$ of Case-3 was characterized by the day-night contrast, manifested as pollution filling the entire
plain area at night while concentrating in front of the mountains with a distinct edge on the southeast side
during the daytime (Fig. 3c, ii-iv). Finally, these pollution cases were diffused in different ways. In Case-
1, the clean air first occupied the northern parts of the NCP with a large concentration gradient on the
front edges (Fig. 3a, v). As for Case-2, $PM_{2.5}$ was restored to a clean level from the northeast (Fig. 3b,
v). Pollution in the northwest was earliest removed in Case-3, with Beijing acting like a
loophole/passageway in the cleaning process (Fig. 3c, v). These cases presented various pollution
distributions, however, all of them were characterized by clear edges or distinct heavy pollution cores.

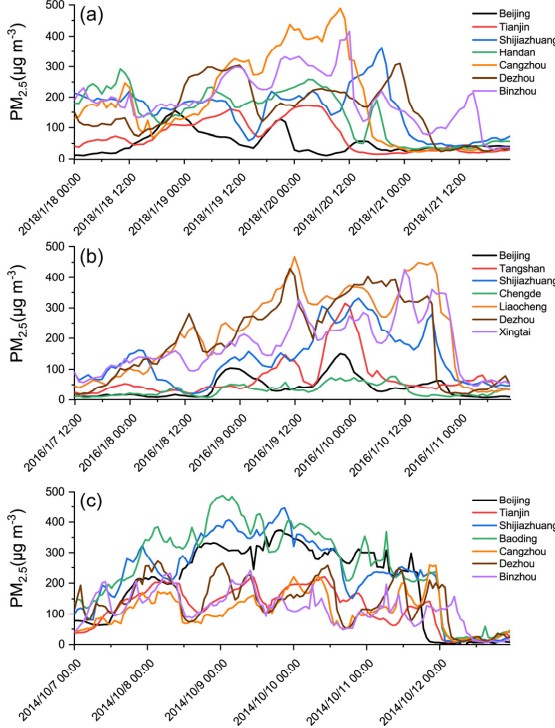


Figure 2. Temporal evolution of $PM_{2.5}$ concentrations during Case1–3, respectively represent (a) west-
southwest wind shear type pollution (January 18–21, 2018), (b) south-north wind shear type pollution
(January 7–11, 2016) and (c) topographic obstruction type pollution (October 7–12, 2014). The locations
of these $PM_{2.5}$ stations are marked in Fig. 1a.
The correspondent surface meteorological fields of the three cases are shown in Fig. 4. Case-1 and
Case-2 represent the two main modes of the wind shear category which are affected by the dynamic AIBs,
and thus the observed sea level pressure and wind fields are discussed (Fig. 4a-b). Case-3 belongs to the
topographic obstruction category affected by the AIB created by the cold-air damming, and its potential
temperature and wind fields are displayed to focus on the combined action of the thermal and dynamic
properties (Fig. 4c). As shown in Fig. 4a, i-iii, the pollution formation and maintenance processes of
Case-1 were dominated by a leeward trough, which induced the westerly airflow shear to the southwest
wind and produced a convergence belt at the trough axis. As the trough broadened and moved eastward,
the wind convergence zone also moved (Fig. 4a, i-iii). On the evening of January 19, 2018, the leeward
trough temporarily evolved into an inverted trough under the force of the approaching high-pressure,



creating a cyclonic convergence (Fig. 4a, iv). This explains why the heavily polluted center transferred
to the east in this episode (refer to Fig. 3a, i-iv). Until January 20, 2018, a high-pressure system invaded
the NCP from the northeast, bringing strong northeast winds (Fig. 4a, v), which made the pollution spread
southward in turn (refer to Fig. 3a, v). During Case-2, a saddle pressure field persisted in the pollution
formation-maintenance stage and induced the prevailing northerly winds in the northern NCP to
antagonize the dominant southerly flows in the southern area (Fig. 4b, i-iv). As a result, the polluted air
mass was prevented from advancing northward to the mountains, causing a strong contrast between the
pollution levels in the northern and southern of the domain (refer to Fig. 3b i-iv). Its pollution diffusion
process was also associated with a northeast high-pressure invasion, by strong northeasterly airflows
cleaning up the $PM_{2.5}$ (Fig. 4b, v). As for the Case-3 under the topographic obstruction category, there
was a narrow area with low potential temperature and weak wind speed at the foot of the mountains on
the windward side in the daytime but this feature became fuzzy at night (Fig. 4c, i-iv). This diurnal
variation repeatedly occurred during the formation and maintenance stage, which may be an important
reason for the day-night difference in pollution distribution (refer to Fig. 3c i-iv). In the end, the strong
flows and cold air bursting like a jet stream through a pathway across Zhangjiakou-Beijing-Tianjin (Fig.
4c, v), made pollutants begin to be swept out from the northwest (refer to Fig. 3c, v).

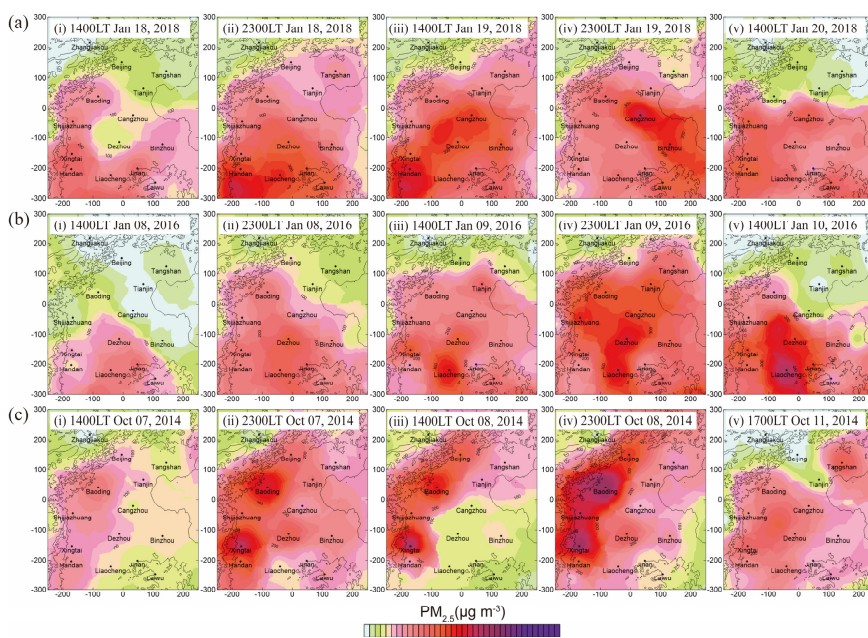


Figure 3. Spatial distributions of surface $PM_{2.5}$ concentrations (shaded colors) at the pollution stages of
(i) formation, (ii-iv) maintenance and (v) diffusion during representative Case1–3 under (a) west-
southwest wind shear type, (b) south-north wind shear type and (c) topographic obstruction type. Values
shown in x- and y-axis denote the distances (km) to the domain center.



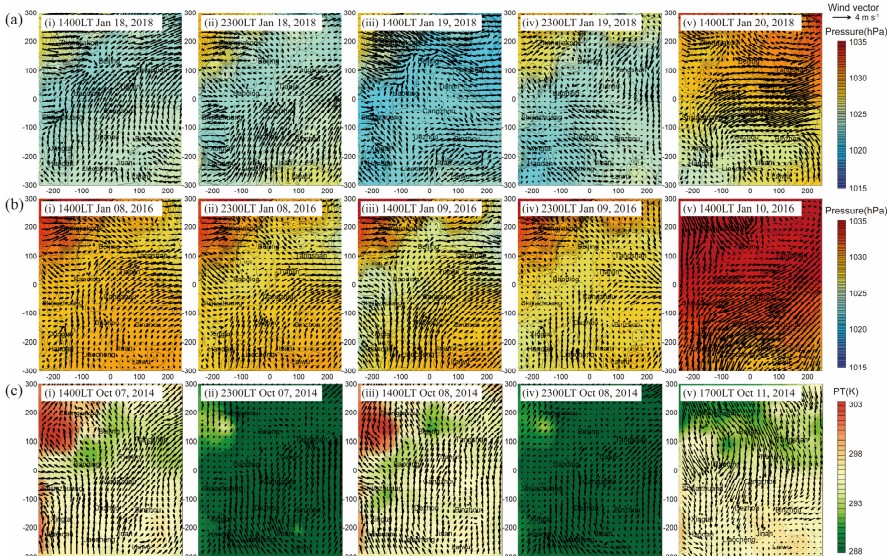

Figure 4. Observed sea level pressure/potential temperature and wind vectors at the pollution stages of (i) formation, (ii–iv) maintenance and (v) diffusion during representative Case1–3 under (a) west-southwest wind shear type, (b) south-north wind shear type and (c) topographic obstruction type. The shaded colors represent the sea level pressure in (a-b) and the potential temperature in (c). The arrows indicate wind vectors. Values shown in x- and y-axis denote the distances (km) to the domain center.

**3.2 Evaluation of simulated meteorological field**

To reveal the PBL three-dimensional structure of these representative cases, numerical simulations were conducted using the WRF model. It is necessary to evaluate the model reliability before analyzing the simulated results. The model-observation comparisons in the previous studies usually focus on the time series of surface meteorological elements, such as 10 m wind speed and direction, 2 m temperature and humidity (Rogers et al., 2013; Bei et al., 2018; Qu et al., 2021). The model performance of their spatial fields is often ignored, and the evaluation of the PBL vertical structure is relatively lacking, but the regional distribution and vertical profile of wind-temperature-humidity are crucial for air pollution. In this study, the evaluation was carried out including three perspectives: i) the temporal evolution and ii) the spatial pattern of near-surface potential temperature and wind speed, as well as iii) the vertical profile-temporal structure of these two variables.

For the temporal evolution of the near-surface potential temperature and wind speed, the hourly observations and simulations of 13 key cities (Beijing, Tianjin, Shijiazhuang, Baoding, Handan, Tangshan, Cangzhou, Dezhou, Jinan, Weifang, Binzhou, Chengde and Zhangjiakou) evenly distributed in the NCP were compared during these three pollution cases. The model outputs were extracted from the grid points nearest to observed sites. As shown in Table1, the correlation coefficients of the simulated





and observed potential temperature and wind speed were 0.80~0.91 and 0.54~0.64 ($p<0.01$), respectively.
The statistical results demonstrated that the major variations in the time series of the surface observations
were reproduced by the model, which has also been recognized in previous studies (Rogers et al., 2013;
Bei et al., 2018; Qu et al., 2021).
Table 1. Statistics of model performance for near-surface potential temperature and 10 m wind speed for
selected 13 cities during the representative cases.

| | Case-1 | | | | Case-2 | | | | Case-3 | | | |
|---|---|---|---|---|---|---|---|---|---|---|---|---|
| | PT | | WS | | PT | | WS | | PT | | WS | |
| | R | RMSE | R | RMSE | R | RMSE | R | RMSE | R | RMSE | R | RMSE |
| Beijing | 0.80 | 2.20 | 0.62 | 1.15 | 0.87 | 2.60 | 0.61 | 1.69 | 0.91 | 2.20 | 0.73 | 1.65 |
| Tianjin | 0.89 | 2.40 | 0.66 | 1.48 | 0.85 | 1.90 | 0.63 | 1.97 | 0.92 | 2.10 | 0.61 | 2.13 |
| Shijiazhuang | 0.77 | 2.80 | 0.52 | 2.02 | 0.82 | 2.50 | 0.66 | 1.69 | 0.88 | 2.20 | 0.58 | 1.95 |
| Baoding | 0.83 | 2.50 | 0.60 | 1.34 | 0.85 | 2.40 | 0.61 | 1.53 | 0.89 | 2.30 | 0.60 | 1.97 |
| Handan | 0.93 | 1.40 | 0.48 | 1.36 | 0.78 | 3.20 | 0.56 | 2.27 | 0.95 | 1.30 | 0.66 | 1.94 |
| Tangshan | 0.69 | 4.00 | 0.62 | 1.44 | 0.81 | 3.30 | 0.53 | 1.64 | 0.85 | 3.00 | 0.46 | 2.24 |
| Cangzhou | 0.85 | 3.00 | 0.64 | 1.23 | 0.79 | 2.50 | 0.60 | 1.92 | 0.94 | 2.10 | 0.75 | 1.45 |
| Dezhou | 0.78 | 3.70 | 0.51 | 1.69 | 0.87 | 1.50 | 0.63 | 2.82 | 0.90 | 2.30 | 0.55 | 2.97 |
| Jinan | 0.76 | 2.80 | 0.49 | 2.96 | 0.74 | 2.40 | 0.63 | 2.45 | 0.91 | 2.10 | 0.56 | 3.10 |
| Weifang | 0.79 | 2.10 | 0.53 | 1.42 | 0.78 | 2.50 | 0.71 | 1.99 | 0.94 | 2.10 | 0.85 | 1.40 |
| Binzhou | 0.81 | 2.50 | 0.51 | 1.97 | 0.83 | 2.30 | 0.86 | 1.29 | 0.92 | 2.00 | 0.81 | 1.47 |
| Chengde | 0.75 | 5.10 | 0.47 | 2.06 | 0.63 | 6.50 | 0.47 | 2.60 | 0.84 | 3.70 | 0.56 | 1.74 |
| Zhangjiakou | 0.90 | 5.40 | 0.33 | 2.23 | 0.77 | 5.30 | 0.47 | 3.13 | 0.96 | 4.80 | 0.54 | 2.50 |
| **Average** | **0.81** | **3.07** | **0.54** | **1.72** | **0.80** | **2.99** | **0.61** | **2.08** | **0.91** | **2.47** | **0.64** | **2.04** |

Case-1: west-southwest wind shear type (January 18–21, 2018); Case-2: south-north wind shear type
(January 7–11, 2016); Case3: topographic obstruction type (October 7–12, 2014). All statistics are
calculated from hourly values.
Referring to Fig. 4, the spatial distribution of the simulated sea level pressure/potential temperature
and wind vector during the three cases are displayed in Fig. 5. In Case-1 and Case-2, the leeward trough
and saddle pressure field, as well as the corresponding west-southwest wind shear and south-north wind
shear were reproduced in the simulated fields (Fig. 5a-b, i-iv vs Fig. 4a-b, i-iv). Also, their movement
and evolution during the pollution formation-maintenance processes were also captured by the WRF
model, although there were some small deviations in the specific positions. At the diffusion stage, the
simulated northeastern high-pressure invasion and the prevailing easterly/northeasterly winds were
comparable with the observed fields (Fig. 5a-b, v vs Fig. 4a-b, v). As for Case-3, the surface
meteorological fields output by the model successfully reflected the narrow cold zone and quiet wind
belt at the foot of the mountains, as well as their diurnal variation and sustainability in the pollution
formation-maintenance stage (Fig. 5c, i-iv vs Fig. 4c, i-iv). Even though the area was shorter at its south
end on the afternoon of October 08, 2014, and there was an overestimate of the potential temperature in
the northwest mountains and the Bohai Sea at night. At the end of this episode, a strong northerly cold
airflow similar to the observation appeared in the simulation field (Fig. 5c, v vs Fig. 4c, v). Generally,
the main features of the surface distributions of meteorological observations during these three cases can
be reflected well in the simulated fields.

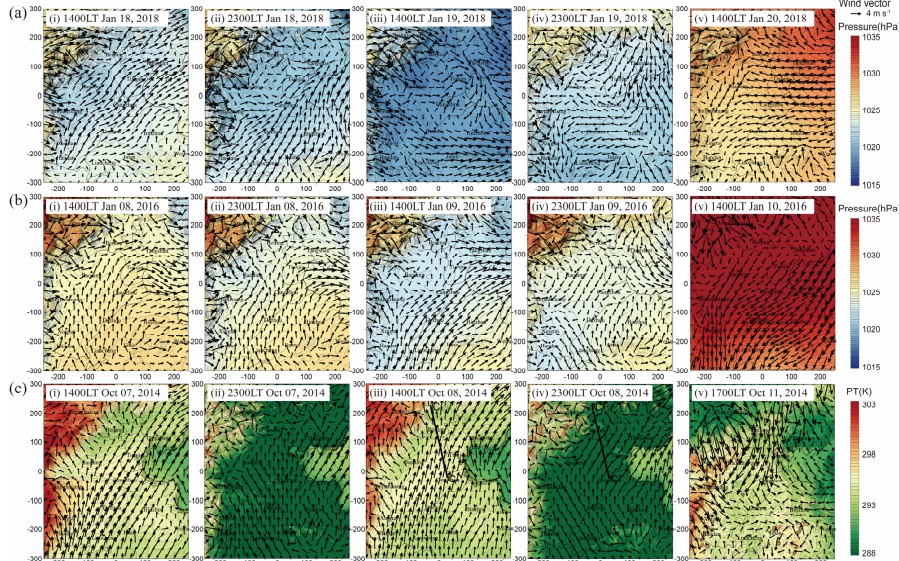


Figure 5. Simulated sea level pressure/potential temperature and wind vectors at the pollution stages of
(i) formation, (ii-iv) maintenance and (v) diffusion during representative Case1–3 under (a) west-
southwest wind shear type, (b) south-north wind shear type and (c) topographic obstruction type. The
shaded colors represent the sea level pressure in (a-b) and the potential temperature in (c). The arrows
indicate wind vectors. Values shown in x- and y-axis denote the distances (km) to the domain center.
Lines $C_1C_1'$ in (c) refer to the cross-sections of the potential temperature in Fig. 11.
Moreover, the simulated and observed time-height cross sections of potential temperature and wind
speed were compared to reveal the model's ability to capture the PBL vertical structure of each case (Fig.
6). The observation data of Case-1 and Case-2 were obtained from intensive sounding experiments at the
Dezhou site and Cangzhou site, respectively. The observation information during Case-3 was provided
by routine soundings at the Beijing site. As for Case-1, the model successfully reproduced thermal
structure evolution in the pollution formation-maintenance period, while the final uplift of the inversion
layer and the growth of PBL were not captured (Fig. 6a, i-ii). By comparison, the dynamic structures,
being the dominant role of this type, were better simulated. The vertical location and temporal transition
of the strong and weak wind layers were comparable with observations (Fig. 6a, iii-iv). The model
performance during Case-2 was satisfactory both for cross-sections of the potential temperature and wind
speed. The formation and decay of upper temperature inversion and the development of the cold

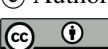

convective PBL were consistent between observation and simulation, though there were some
underestimations in the modeled results (Fig. 6b, i-ii). The weak wind layer presented in the maintenance
stage and vertical wind shear that occurred in the diffusion stage were also captured by the model with
smaller gradients (Fig. 6b, iii-iv). In Case-3, the WRF reproduced the observed diurnal cycle of the
potential temperature in the low-level and the continuous warming at the upper layer during the
formation-maintenance process, as well as the replacement of a well-mixed cold air mass in the last phase
(Fig. 6c, i-ii). The evolution of the simulated wind speed was roughly similar to the observation,
including the maintenance of the calm wind layer in the first four days and the appearance of the final
strong wind layer (Fig. 6c, iii-iv). There were some inconsistencies in the details of observation and
simulation evolution, which may result from the coarse resolution of routine soundings in time and
vertical direction, in addition to the uncertainties of model simulation.

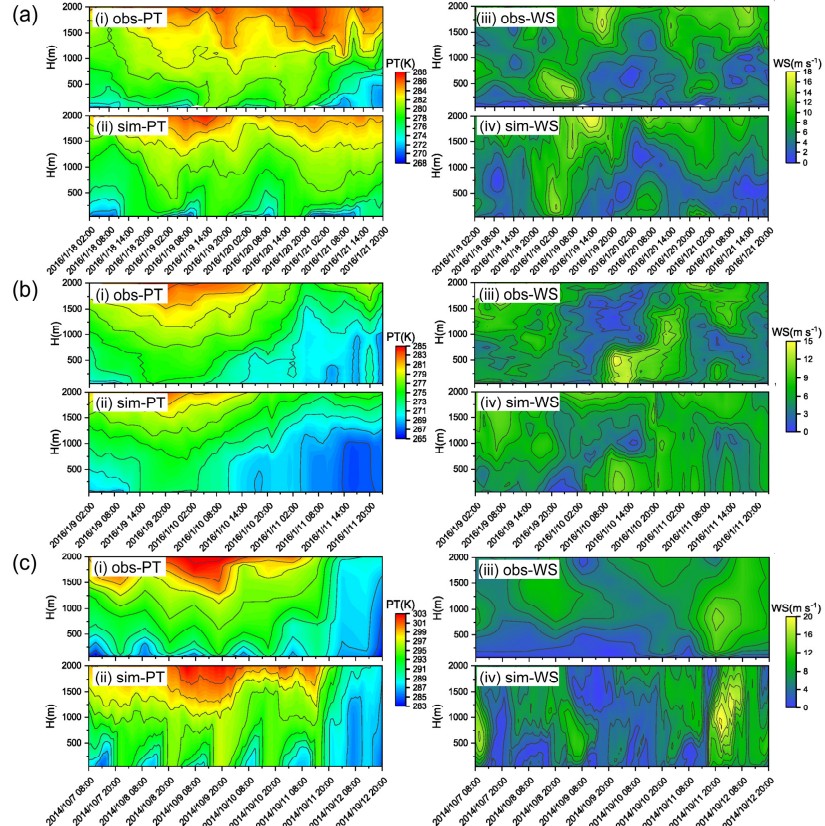


Figure 6. Observed and simulated time-height cross sections of potential temperature (left) and wind
speed (right) during representative Case1–3 under (a) west-southwest wind shear type (January 18–21,
2018), (b) south-north wind shear type (January 9–11, 2016) and (c) topographic obstruction type
(October 7–12, 2014).



Overall, the model shows the ability to capture the observed mesoscale systems and atmospheric
thermal-dynamic structures reasonably both at the surface and in the vertical direction. With confidence
in the model results, we now proceed to a detailed investigation of the PBL spatial structure affected by
mesoscale AIBs under various pollution types.
**3.3 PBL spatial structure under each pollution type**
We analyze the simulated vertical cross-sections of the mesoscale systems and AIBs to reveal the
three-dimensional structure of the PBL. Two key parameters, potential temperature and wind divergence,
are used to respectively indicate the thermal stability and dynamic convergence of the PBL, which affect
the vertical mixing and horizontal diffusion of $PM_{2.5}$, and are critical to the pollution formation and
distribution. Therefore, for the pollution case under the thermally dominated frontal category, dynamical-
driven wind shear category, and the thermodynamic-mixture topographic obstruction category, the
potential temperature section, wind divergence section and both of them are respectively displayed and
discussed.

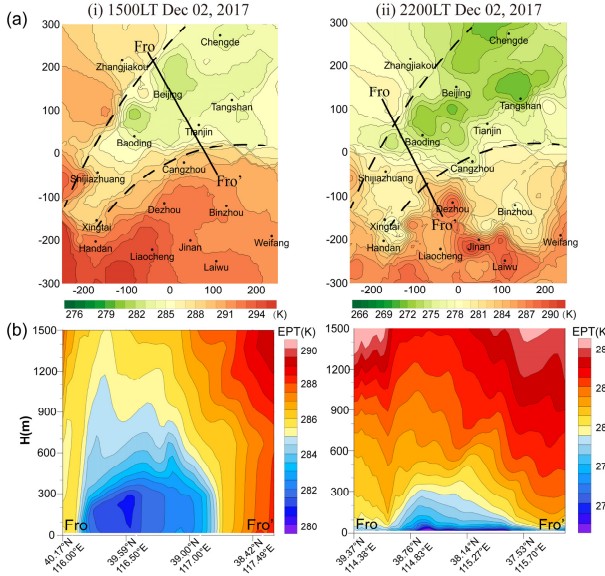


Figure 7. Re-display of Figs. 5&7 in Jin et al. (2021). (a) Surface distributions and (b) vertical cross-
sections of equivalent potential temperature at (i) 1500 LT and (ii) 2200 LT on December 2, 2017, in
frontal category case. Dashed and solid lines in (a) respectively indicate the locations of the AIBs and
the sections.
**Frontal category**
The three-dimensional thermal structure of the PBL under the frontal category has been revealed in



a previous case study (Jin et al., 2021). Statistics show that this kind of mesoscale PBL structure occurs
most frequently and tends to result in the most severe pollution levels in the NCP, so we recapitulate it
again here. As shown in Fig.7 ( the re-display of Figs. 5&8 in Jin et al. (2021)), the boundary layer was
characterized by an isolated cold air mass, which was laterally confined by mountains and warm front
AIB (Fig.7a-b), and vertically covered by a warm dome (Fig.7c-d). The elevated inversion strength was
as high as 3~6 K 100 m$^{-1}$, making the PBL height drop abruptly to 200~300 m in the cold area from
600~800 m outside the zone (Fig.7c). The contrast of the PBL thermal structure was unobvious during
nighttime, with surface inversion over the whole region (Fig.7d). However, the nocturnal inversion layer
was thicker and stronger in the cold area, making the PBL height lower than in the warmer area. The
shallow stable stratified PBL structure persisted throughout the daytime and night, which constituted
adverse dispersion conditions and further resulted in the most serious PM$_{2.5}$ pollution.
**Wind shear category**

This pollution category, consisting of three subtypes of west-southwest wind shear, southeast-east

wind shear and south-north wind shear, is mainly driven by dynamic flows. Therefore, the wind
divergence features, including near-surface horizontal distributions, vertical cross-sections and vertical
profiles are considered. Among these subtypes, the dynamic characteristics of the first two are similar,
so we only analyze the representative cases of the first and the third subtypes, in the following.

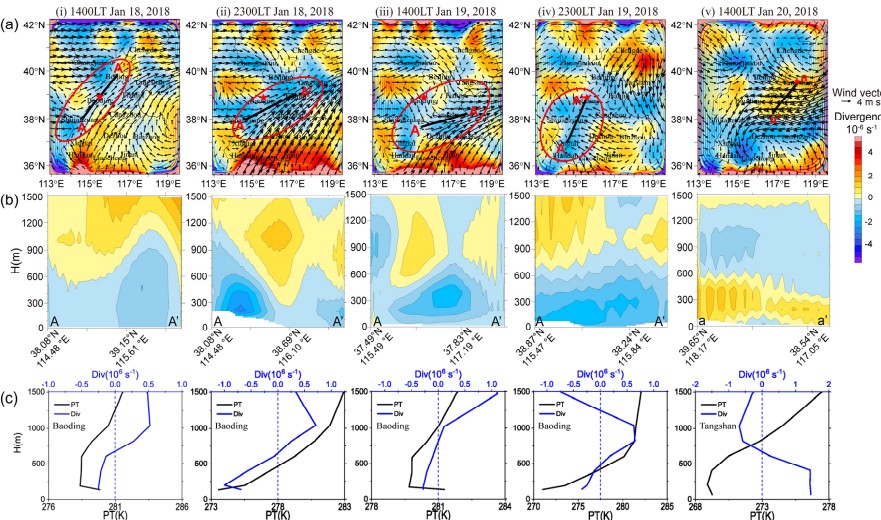

Figure 8. (a) Surface spatial distributions, (b) vertical cross-sections and (c) vertical profiles of the
simulated wind divergence at the pollution stages of (i) formation, (ii-iv) maintenance and (v) diffusion
during representative Case-1 under west-southwest wind shear type. The red ellipses, black lines and red
pentacles in (a) indicate the convergence belt, the section lines in (b) and the profile sites in (c),
respectively. The potential temperature profiles are presented in (c) to indicate the boundary layer top.





Figure 8 displays the PBL dynamic structure of Case-1 which belongs to the west-southwest wind
shear type. During the pollution formation-maintenance stage, westerly winds shifted to southwesterly
winds at the trough axis and thus formed a convergence belt at the surface with a divergence of
$-2\sim-4\times10^{-6}$ s$^{-1}$. This trough-convergence belt continued to move to the southeast, and evolved into a
cyclonic-convergence center at the end of the maintenance phase (Fig. 8a, i-iv). The vertical sections of
the surface convergence belt show that the depth of the convergence layers did not exceed 1000 m, with
compensating divergence layer immediately above it (Fig. 8b, i-iv). Furthermore, the vertical profiles of
the wind divergence and potential temperature at the Baoding site located in the convergence belt were
extracted to illustrate PBL dynamic structure more clearly. It can be found that the mutation of divergence
value and the jump of potential temperature roughly appeared at the same height (Fig. 8c, i-iv), which
demonstrated the vertical scale of the wind convergence belt was equivalent to the depth of the PBL.
This phenomenon reveals that the west-southwest wind convergence caused by the trough mainly occurs
within the PBL, reflecting its mesoscale property. In the process of pollution diffusion, divergent wind
fields first occurred in the northeastern area (Fig. 8a, v). The vertical cross-section of this divergent layer
and vertical profiles at the Tangshan site show that the northeast wind divergence layer was relatively
thin with a thickness of no more than 600 m (Fig. 8b-c, v), meaning that the removal of pollutants only
occurred in the low-level atmosphere.

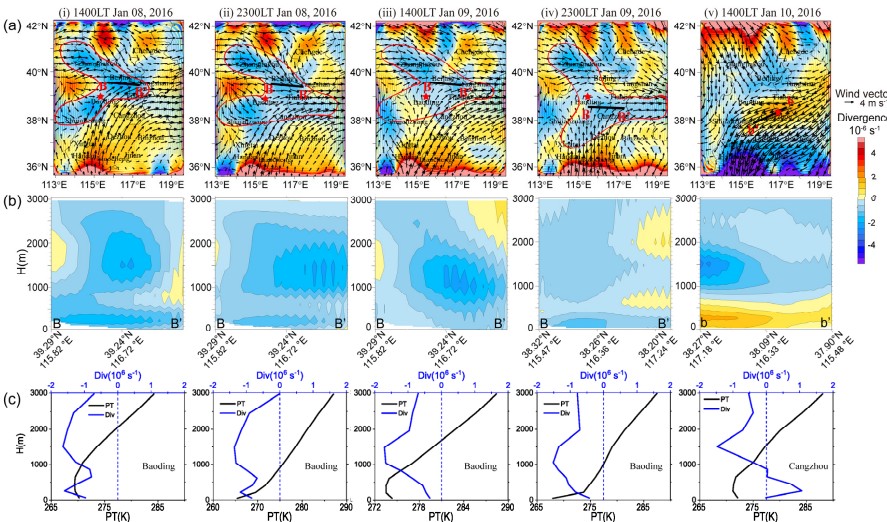


Figure 9. Same as Fig. 8, but for representative Case-2 under south-north wind shear type. The red lying-

Y shapes, black lines and red pentacles in (a) indicate the convergence belt, the section lines in (b) and

the profile sites in (c), respectively.

As for the south-north wind shear type, the surface divergence fields displayed a "lying Y shaped"
convergence zone with the opening to the left during the pollution formation-maintenance stage of the



representative Case-2 (Fig. 9a, i-iv), which was caused by the meeting of the southerly winds and the
northerly winds and then turning to the easterly winds. The vertical cross-sections of this special
convergence zone exhibited a depth extending upwards for more than 3000 m, with a peak between 1000
m and 2000 m (Fig. 9b, i-iv). With reference to the vertical profiles of wind divergence and potential
temperature at the Baoding site, it can be seen that the depth of the convergence layer under this shear
type far exceeded the height of the PBL, whether it was in the daytime or the night (Fig. 9c, i-iv). These
phenomena prove that the south-north wind shear created by the saddle pressure field was exceeded the
scale of a typical mesoscale meteorological process. The vertical scale of the dynamic feature was no
longer limited to the PBL, implying the sub-synoptic scale characteristics. In the pollution diffusion stage
of this case, the PBL structure was the same as in Case-1 (Fig. 9a-c, v), and has been described in the last
paragraph.
**Topographic obstruction category**
As an outcome of a mixture of thermal and dynamic effects, the topographic obstruction category
pollution is analyzed from the perspectives of both the wind divergence and potential temperature to
reveal the thermal and dynamic structure of the PBL.

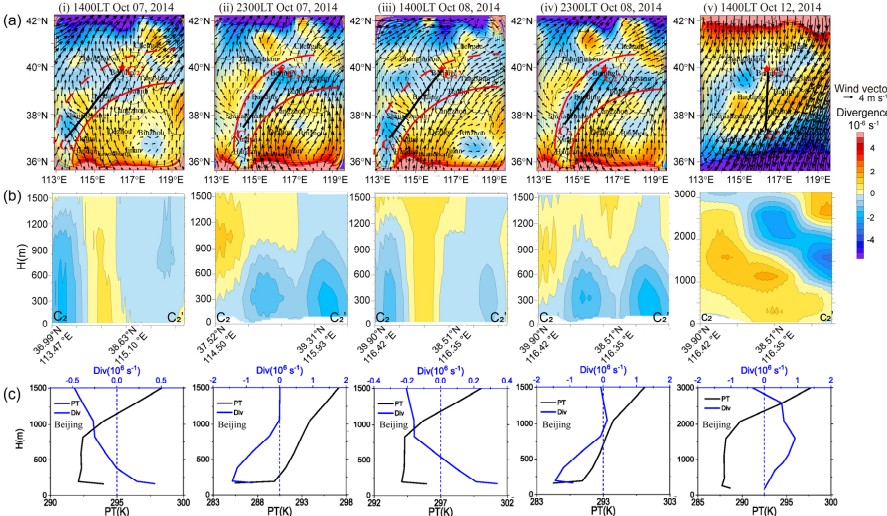


Figure 10. Same as Fig. 8, but for representative Case-3 under topographic obstruction type. The red
curves, black lines and red pentacles in (a) indicate the convergence belt, the section lines in (b) and the
profile sites in (c), respectively.
Figure 10 shows the dynamic characteristics of the PBL during the representative Case-3. In the
pollution formation-maintenance stage, there was an arc-shaped convergence belt at the foot of the
mountains on the windward, due to the momentum loss in the northward flow under the action of
topographic obstruction (Fig. 10a, i-iv). The shape of this convergent belt was more regular at night (Fig.



10a, ii, iv) but had some breakages at the northwest edge during the day when there was a local southeast
wind around Beijing and Shijiazhuang (Fig. 10a, i, iii). The vertical sections also reflected the general
features and diurnal difference, manifested as an integral convergence layer with the depth approximating
the height of the mountain at night (Fig. 10b, ii, iv), and an isolated divergent layer emerged during the
daytime (Fig. 10b, i, iii). The vertical profiles of the wind divergence and potential temperature at Beijing
were further extracted shown in Fig. 10c, i-iv. In the evening, the atmosphere below 1200 m was
convergent with the peak appearing near the surface about $-1.5 \times 10^{-6}$ s$^{-1}$. In the afternoon, there was a
weak divergence layer with a strength of about $0.5 \times 10^{-6}$ s$^{-1}$ and a thickness of about 200~300 m within
the PBL. We infer that the day-night variation may be the consequence of the mountain-valley circulation,
where northwestward daytime valley winds developed along the mountain passes near Beijing and
Shijiazhuang leading to flow divergence, and downslope winds formed at night strengthening the surface
wind convergence. When the pollution began to spread, the northern part of the domain was in a strong
divergence condition (Fig. 10a, v). The corresponding cross-section shows that the north wind divergence
layer was very deep (nearly 3000 m), gradually thinning from north to south (Fig. 10b, v). Moreover, the
vertical profiles of the divergence and potential temperature at the Beijing site prove that the PBL was
well developed up to 2000 m, accompanied by strong horizontal divergence throughout the layer (Fig.
10c, v), both of which indicate extremely favorable ventilation conditions.
The thermal properties and their evolution, especially diurnal variation, play an important role in
this type of pollution pattern, which has been presented in the previous surface analysis. Hence, we
further explore the three-dimension thermal structure of the PBL, taking the vertical cross-sections of
potential temperature across the characteristic cold area in the pollution maintenance stage (October 8,
2014, the location of cross-section shown in Fig. 5c) as an illustration. In the early hours of the morning,
although there were surface inversions, the cold air masses in front of the mountains were much thicker
(Fig. 11a, i). After sunrise, the convective boundary layer developed both in the front of the mountains
and in the plain due to the surface heating, but the temperature in the southern plain was higher (Fig. 11a,
ii). In the afternoon, a deep, well-mixed warm PBL has formed in the southern plains while a cold air
mass capped by strong inversion still remained in the northern piedmont area (Fig. 11a, iii). At night,
large amounts of cold air accumulated at the foot of the mountains again (Fig. 11a, iv). The vertical
profiles of the simulated potential temperature of the three cities from south to north, Jinan, Cangzhou
and Beijing, also support this thermal evolution process. At 0200 LT, there were surface inversions at all
three cities, and Beijing had the strongest inversion intensity of about 2 K 100 m$^{-1}$ (Fig. 11b, i). By 1000
LT, the PBL in Jinan had increased to 1100 m, while the convective boundary layers in Beijing and
Cangzhou were shallow (about 400 m, Fig. 11b, ii). In the afternoon, the PBL was fully developed with
the height from the south to the north site ranging from 1150 m to 650 m, and there was still a thick
inversion layer above Beijing (Fig. 11b, iii). At 2300 LT, the surface inversion at the three sites has
formed again (Fig. 11b, iv). The persistent cold air mass in front of the mountains is similar to the cold-
air damming on the eastern side of Appalachian (Bell and Bosart, 1988). The prevailing southerly warm
airflows were blocked by the mountains and the geostrophic balance was disrupted so that the heat cannot
reach the foothills and the air further accumulated and ascended here for adiabatic cooling. It should be
noted that the southeast edge of this cold area was more pronounced during the daytime (Fig. 4c, Fig.
11), in comparison to that at the night. This is reasonable given that the nocturnal boundary layer was
stable over the whole domain and more susceptible to the local property. Although the AIB was relatively
unclear at the surface during nighttime, the nocturnal cold layer at the foothills was deeper than the
southern plain area, probably due to the cold drainage flows along the sidewall of the mountains forming
a cold air pool. This diurnal cycle of the PBL thermal structure can well explain the day-night difference
in pollution distribution pattern (Fig. 3e). Shortly speaking, the PBL thermal structure during the
formation-maintenance stage under the topographic obstruction category pollution is a manifestation of
cold-air damming AIB, assisted by the role of cold air pool at night (Fig. 4c).

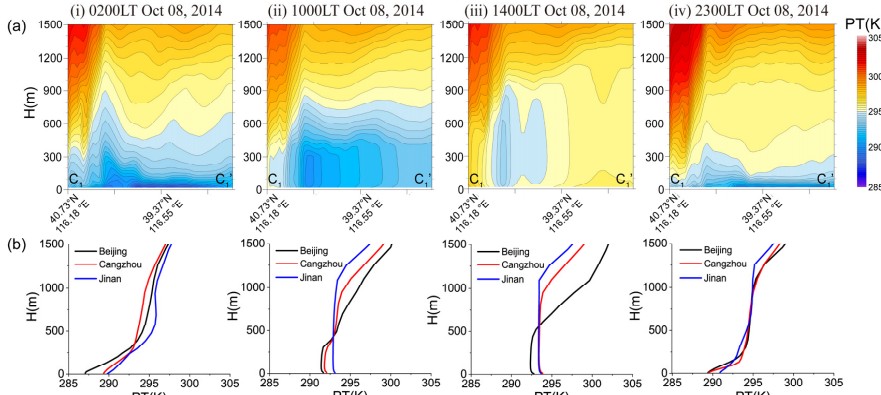


Figure 11. (a) Vertical cross-sections and (b) vertical profiles of the simulated potential temperature at
(i) 0200 LT, (ii) 1000 LT, (iii) 1400 LT and (iv) 2300 LT on October 08, 2014 in Case-3 under the
topographic obstruction type. The cross-sections $C_1C_1'$ are shown Fig. 5c, iii, iv.
**4 Summary and discussion**
The three-dimensional PBL structures modified by the mesoscale systems and interacted with the
AIBs under various pollution types in the NCP, as well as the ability of the mesoscale meteorological
model (WRF) in simulating these processes, were investigated in this study. The pollution types were
classified from pollution episodes during autumn and winter of 7 years (2014–2020) in Jin et al. (2022
submitted). Representative cases under these types were simulated by the WRF model in this study. The
model was comprehensively evaluated for its reliability, by comparison with observed PBL vertical
structure, as well as the temporal series and spatial evolution of the surface meteorological fields.

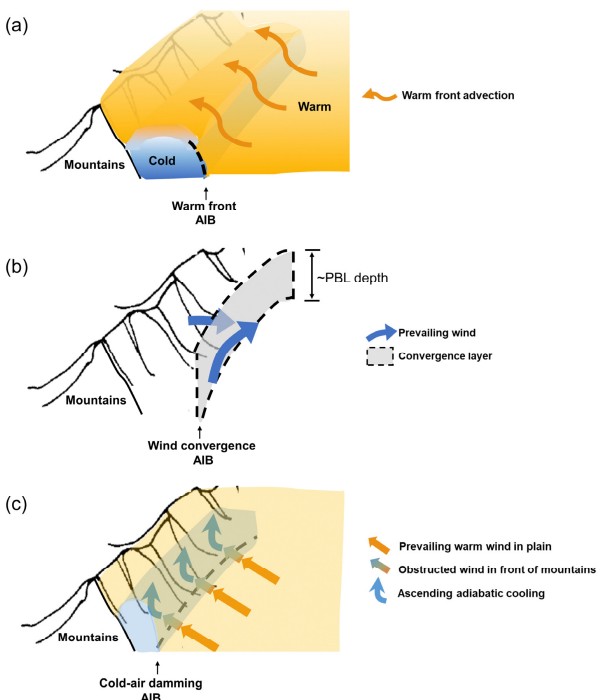


Figure 12. Schematic diagram showing the conceptual model of PBL spatial structures under three
categories pollution. (a) Frontal category: the blue-shaded and orange-filled areas represent the isolated
and stable cold air mass in front of the front and the warm well-mixing atmosphere behind the front. The
orange arrows represent warm front advection. (b) Wind shear category: two blue arrows represent the
airflows ahead of and behind the trough. The gray-filled area indicates the dynamic convergence layer
equal in height to the boundary layer. (c) Topographic obstruction category: the light blue filled area
indicates the cold-air damming at the foot of the windward mountains. It is the result of the regional
warm airflows (long orange arrows) being blocked by topography (short gradient-color arrows) and then
accumulating to ascend cooling (up blue arrows). Black dashed lines in (a-c) indicate the warm front
AIB, wind convergence AIB, and cold-air damming AIB, respectively.

Based on results of this paper, more complete and clearer view of the PBL spatial structures during

pollution episodes in the regional scale of NCP can be obtained, as schematically shown in Fig.12. All
the pollution conditions were classified into three categories. The most prominent was the frontal
category. With an isolated cold air mass laterally bounded by the warm frontal AIB in one side and
mountains in another side, the PBL was vertically suppressed by a dome-like warm cap. Typically, the
intensity of the frontal inversion can be as large as 3~6 K 100 m$^{-1}$. As a consequence, the PBL in this
cold area was very shallow (as low as 200~300 m) and kept stable stratification, in sharp contrast to the



deep and well-mixing boundary layer outside this zone (Fig. 12a). This explained why $PM_{2.5}$ accumulated
rapidly in this enclosed and stable space and formed a laterally clearly defined polluted air mass.
Diurnally, the nocturnal PBL in this category was less typical as its daytime counterpart. The thermal
structure of the PBL played a leading role in this category, resulting in the most severe pollution level.

The second category was dominated by dynamic processes. Three modes compounded this category,

i.e., west-southwest wind shear type, southeast-east wind shear type and south-north wind shear type.
The first two types were characterized by the leeward trough (or inverted trough). A convergence layer
lay in the wind shear zone with the thickness of the PBL depth (Fig. 12b), with a typical near-surface
divergence of $-2 \sim -4 \times 10^{-6}$ s$^{-1}$, and accompanied by a compensating divergence layer above the PBL,
which reflected the mesoscale property of the trough AIB. The third type displayed a "lying Y shaped"
convergence layer from the surface extending upwards to about 3000 m, with a convergence peak above
the PBL top (not shown in Fig. 12). This implied the sub-synoptic scale features. In this category, the
boundary layer was dominated by dynamic convergence effects, which made pollutants accumulate, and
the pollution level in the NCP was relatively light.

The topographic obstruction pollution category was characterized by a cold-air damming AIB at the

foot of the windward side of the mountains. It usually occurred when the southerly winds were too weak
to cross the terrain barrier and the northward flows were blocked, which allowed air masses to accumulate
and ascend cooling in front of the mountains. The PBL air was cold and capped by a strong inversion in
the damming area, in contrast with well-mixed warm PBL in the southern plains. Meanwhile, the air
flows were convergent in front of the mountains. These general characterizes are shown in Fig. 12c. In
more detail, the thermal discontinuity became indistinct at night due to the surface inversion over the
whole domain, while the nocturnal wind convergence belt was more pronounced. The diurnal variation
of the PBL dynamic and thermal structure made the pollutants concentrate at the foot of the mountains
during the daytime and distribute throughout the entire plain at night.

It should be emphasized that the above results are highly dependent on numerical simulation, due

to the scarcity and limitation of PBL sounding data. Evaluation from the spatial-temporal variation of the
surface meteorological field and PBL vertical structure indicates that the model performance was good.
WRF can capture mesoscale systems and AIBs, as well as their overall evolution process and diurnal
variation. However, it was still difficult to reproduce the precise timing of the buildup and breakup as
well the exact location and range of these systems. This deficiency should be concerned seriously when
simulated meteorological fields are used to drive air quality models, since a small position bias and time
deviation of the AIBs can significantly alter pollution levels at a certain site (Seaman, 2000; McNider
and Pour-Biazar, 2020). In the present study, we focus on the major characteristics of the PBL spatial
structure, which were reasonably reflected in the model results.

In addition to the inevitable uncertainty of the numerical simulation, the classification proposed in



this paper might still be roughly or oversimplified. Real processes may be more complex and atypical as
analyzed in this paper. However, this work, to the authors' knowledge, is the first trial to classify the PBL
structure over the vast scale of the NCP, and to clarify its role on regional $PM_{2.5}$ pollution. Modulation
of the PBL by mesoscale meteorological processes, particularly the AIBs, is emphasized. Extending the
view of the PBL from local vertical properties to mesoscale three-dimension structures may be a step
toward a better understanding of the meteorological effects on regional-scale $PM_{2.5}$ pollution.

**Data availability**

The data in this study are available from the corresponding author (xhcai@pku.edu.cn).

**Author contribution**

XHC and XPJ designed the research. MYY and HSZ collected the data. XPJ performed the simulations
and wrote the paper. XHC reviewed and commented on the paper. YS, XSW and TZ participated in the
discussion of the article.

**Competing interests**

The authors declare that they have no conflict of interest.

**Acknowledgements**

This work was supported by National Key Research and Development Program of China
(2018YFC0213204).

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
