# Peer review of "Regional PM2.5 pollution confined by atmospheric internal boundaries in the North China Plain: boundary layer structures and numerical simulation"

_Atmospheric Chemistry and Physics, 2022_

## Author Response (AR1)

**Response to Reviewers**

**Manuscript:** Regional PM$_{2.5}$ pollution confined by atmospheric internal boundaries in the North China Plain: boundary layer structures and numerical simulation (acp-2022-48)

**Response to Reviewer #1:**

We are grateful for the positive evaluation and comment from Reviewer #1. Based on these suggestions, we have carefully revised the manuscript. There are three major changes: (1) The reasons for using the WRF model rather than the WRF-Chem model are discussed; (2) The PBL height evolution in both observation and simulation is presented and analyzed; (3) The vertical cross-sections of potential temperature during Case-1 and Case-2 are added in the supplementary material. In addition, the manuscript structure is improved as suggested by Reviewer #2, by moving Fig.12 and related descriptions into the Introduction, and deleting Fig. 7 and the paragraph "Frontal category". The numbering of all figures is changed accordingly.

The response to each comment is listed below. The original comments are in *blue and italic*, our replies are in normal font. Bracketed numbers are used for referee comments (e.g., *[R1.1]*).

**Summary**

*The manuscript investigated the three-dimensional PBL structures under various pollution types in the North China Plain (NCP) by using the WRF model during autumn and winter of seven years (2014-2020). They proposed three pollution PBL types (frontal category, wind shear category, and topographic obstruction pollution category) and investigated the two main types of wind shear category and topographic obstruction category through case studies in this paper. Such work is a good supplement to the synoptic-scale and boundary-layer scale studies and I believe it will be of interest to the community of atmospheric pollution and boundary-layer meteorology. Overall, the paper is logically structured and well written. However, some details and explanations on methods and data need to provide to justify and support the conclusions. Thereby, I suggest a major revision before the paper can be accepted by Atmospheric Chemistry and Physics. My detailed comments are listed below.*

**Response:** We thank the reviewer for the positive evaluation and comment on this manuscript. The manuscript has been revised accordingly.

**Major comments**

*[R1.1] My first concern is about the use of the WRF model. Given this study is focused on the different types of aerosol pollution cases, then using the atmospheric chemical transport model WRF-Chem or WRF-CMAQ to simulate these pollution events sounds more plausible than the pure-meteorological model WRF. Otherwise, it is difficult to convince the reader that these pollution cases are reasonably captured without evaluating the performance of simulating PM$_{2.5}$ concentration. Moreover, the interaction between aerosols and boundary layer can modify the PBL thermal and dynamic structures, so I wonder if the pure-meteorological model WRF is suitable for investigating such pollution cases. The authors at least give some discussions on this.*

**Response:** We thank the reviewer for this very critical comment. We have adopted the suggestion and added an explanation to our research method in the revised manuscript. The reasons for using only the WRF model are as follows.

Generally, air pollutant emission, meteorological condition and chemistry, three of them are determinants of air pollution. The present study focuses on the mesoscale PBL structures under pollution conditions, and we try to combine the strength/advantage of both observation facts and numerical simulation capacity to investigate this issue. The densely distributed network of monitoring stations provides reliable PM$_{2.5}$ pollution facts. WRF model provides boundary layer meteorology information. A chemical transport model (e.g., WRF-Chem or WRF-CMAQ) can of course simulate the whole process from meteorology to pollutants transport/diffusion and chemical transformation. But the uncertainties caused by emission inventory, chemical mechanism, and meteorology conditions may complicate the simulation results together (e.g., Travis et al., 2016; Bouarar et al., 2019; Wang et al., 2021). Our current work isolates the boundary layer meteorology simulation from other factors, and evaluates the model intensively by observation data. Therefore, this study relies more on observations, but utilizes ultimately the capacity of the meteorological model to reveal the three-dimensional structure of PBL when pollution occurs.

We admit that, by the present method, the interaction between aerosols and boundary layer properties cannot be analyzed. However, the modification of aerosols to the PBL structures is a special issue to be discussed. The present study focuses on the more traditional part: the impact of meteorology on PM$_{2.5}$ pollution.

The relevant discussion added in Lines 612-630 of the revised manuscript is presented below.

"The present study focuses on the characteristic mesoscale PBL structures under pollution conditions, and emphasizes their role in shaping regional pollution patterns. The analysis of pollution evolution is based on the PM$_{2.5}$ concentration fields interpolated or diagnosed from monitoring data, relying on densely distributed stations. However, the PBL spatial structure is presented by numerical simulation, due to the scarcity and limitation of sounding data. Evaluation from the spatial-temporal variation of the surface meteorological field and PBL vertical structure indicates that the model performance is good. WRF can capture mesoscale systems and AIBs, as well as their

overall evolution process and diurnal variation. It should be noted that, it is still difficult to reproduce the precise timing of the buildup and breakup as well as the exact location and range of these systems. This deficiency should be concerned seriously when simulated meteorological fields are used to drive air quality models, since a small position bias and time deviation of the AIBs can significantly alter pollution levels at a certain site (Seaman, 2000; McNider and Pour-Biazar, 2020). Accurate capture of mesoscale AIBs is a necessary prerequisite for reliable simulation of pollution evolution. Besides, successful reproduction and forecast of air quality by the chemical transport models also involves other factors, such as the accuracy of source inventories and the complexity of chemical mechanisms, which are beyond the scope of this study. The aim of the present work is to provide a clear cognition of these typical PBL structures reproduced by numerical simulations. This goal is achieved satisfactorily."

*[R1.2] Since PBL height is a key parameter in characterizing the PBL structures and the pollution formation mechanism, I am afraid that the authors fail to present this diagnosed variable in the whole manuscript, either for simulation or observation. Moreover, I suggest the author present the vertical cross-sections of potential temperature (like Figure 7) for wind shear category and topographic obstruction category in the supplementary materials to justify that those cases do not belong to the first frontal category.*

**Response:** We thank the reviewer for these suggestions. The PBL height has been presented and analyzed in the revised manuscript, including the comparison between the observations and simulations in Lines 367-370, 372-373, 379-382 and 387-389, and its relationship with wind and temperature structures in Lines 400-401, 441-443, 457-459, 468-469, 504-507 and 517-521. Meanwhile, the PBL height has been indicated in the vertical cross-sections of wind divergence and potential temperature (new Figs. 7-11) to better support the above explanations.

    The vertical cross-sections of potential temperature during Case-1 and Case-2 have been shown in the supplementary material, which illustrates that there is no significant discontinuity in the atmospheric thermal structure and demonstrates their non-frontal characteristics.

    For your convenience, the added sentences on describing the PBL height and the supplementary cross-sections of potential temperature (Figs. S2-S3) are presented below.

    "As for Case-1, the model successfully reproduces the thermal structure evolution in the pollution formation-maintenance period, while the final uplift of the inversion layer and the growth of PBL are not well captured with an underestimation of about 200-300 m. The correlation coefficient (R) between simulated and observed PBL height is about 0.68 (p<0.01). During Case-2, observed and simulated PBL heights show a consistent evolution with a correlation coefficient as high as 0.78 (p<0.01). Both of their PBL heights are lower during the pollution formation-maintenance stage and increase by more than 1000 m in the diffusion stage. In Case-3, the PBL height is characterized by typical diurnal variations during the polluted period, and begins to

abruptly develop in the evening of October 12, 2014, associated with the cold air mass and strong wind, both in observation and simulation (R=0.81, p<0.01)."

"During the pollution formation-maintenance stage of Case-1, the vertical section across the surface convergence belt shows that the depth of the convergence layer did not exceed 1000 m, with a compensating divergence layer immediately above it, being consistent with the evolution of the PBL (Fig. 8b, i-iv). In the process of pollution diffusion, a northeast wind divergence layer was relatively thin with a thickness of no more than 600 m (Fig. 8b-c, v), implying that the removal of pollutants only occurred within the PBL. As for the south-north wind shear mode, the vertical cross-sections of this special convergence zone exhibited a depth extending upwards for more than 3000 m, with a peak between 1000 m and 2000 m above the PBL top (Fig. 9b, i-iv). During Case-3, a deep, well-mixed warm PBL (with a height of more than 1000 m) has formed in the southern plain while a cold air mass capped by strong inversion (at the height of about 600-1000 m) still remained in the northern piedmont area in the afternoon."

[Figure]

Figure S2. (a) Surface spatial distributions and (b) vertical cross-sections of the simulated potential temperature at the pollution stages of (i) formation, (ii-iv) maintenance, and (v) diffusion during representative Case-1 under west-southwest wind shear mode. The black lines in (a) indicate the section lines in (b). The purple dashed lines in (b) indicate the PBL heights.

[Figure]

Figure S3. Same as Fig. S2, but for representative Case-2 under south-north wind shear mode.

**Minor comments**

*[R1.3] Lines 1-2. I suggest the author give the study period in this sentence, otherwise, this sentence will be inaccurate.*

**Response:** Accepted. The study period has been added in Line 2 of the revised manuscript.

*[R1.4] Line 36. Should be "Petäjä et al., 2016".*

**Response:** "Petaja, 2016" has been corrected to "Petäjä et al., 2016" in Line 56 of the revised manuscript.

*[R1.5] Line 100. Should be "from December 25, 2017, to January 24, 2018".*

**Response:** We are sorry for this mistake. It has been corrected in Line 171 of the revised manuscript.

*[R1.6] Line 103. Please give the full name of LT and UTC when they appeared for the first time.*

**Response:** The full names of LT and UTC (i.e., Local Time = Universal Time Coordinated + 8) have been given in Lines 174-175 of the revised manuscript.

*[R1.7] Lines 107-108. What does the original data mean here?*

**Response:** The original data refers to raw data directly measured by radiosonde soundings, which has higher vertical resolution than publicly available data from the Wyoming University, USA (http://weather.uwyo.edu.html). The related explanation has been added in Lines 179-181 of the revised manuscript.

**[R1.8]** *Lines 115-116. Could you show the comparison of the observational and simulated PBL depth?*

**Response:** The observational and simulated PBL heights during these pollution episodes have been superimposed on new Fig. 7 for comparison, and the relevant descriptions are added in the revised manuscript in Lines 367-370, 372-373, 379-382 and 387-389, as follows.

"As for Case-1, the model successfully reproduces the thermal structure evolution in the pollution formation-maintenance period, while the final uplift of the inversion layer and the growth of PBL are not well captured with an underestimation of about 200-300 m. The correlation coefficient (R) between simulated and observed PBL height is about 0.68 (p<0.01). During Case-2, observed and simulated PBL heights show a consistent evolution with a correlation coefficient as high as 0.78 (p<0.01). Both of their PBL heights are lower during the pollution formation-maintenance stage and increase by more than 1000 m in the diffusion stage. In Case-3, the PBL height is characterized by typical diurnal variations during the polluted period, and begins to abruptly develop in the evening of October 12, 2014, associated with the cold air mass and strong wind, both in observation and simulation (R=0.81, p<0.01)."

**[R1.9]** *Lines 131-133. I wonder if the vertical grid resolution is enough to resolve the PBL structure. It is better to give the detailed height of the model level within 2 km?*

**Response:** The detailed heights of the model level within 2 km are as follows: 9 m, 25 m, 50 m, 85 m, 120 m, 160 m, 200 m, 240 m, 290 m, 350 m, 420 m, 500 m, 580 m, 660 m, 740 m, 820 m, 900 m, 980 m, 1080 m, 1200 m, 1350 m, 1550 m, 1700m, 1850 m, and 2000 m. They are listed in Section 2.2 in Lines 205-208 of the revised manuscript.

**[R1.10]** *Line 148. The authors do not mention this study period in section 2.1.*

**Response:** The study period of Case-3 (October 7–12, 2014) corresponds to the routine radiosonde sounding period, which has been stated in Lines 177-179 of the revised manuscript.

**[R1.11]** *Figure 2. I am wondering why the authors present PM$_{2.5}$ concentrations at different sites for these three cases?*

**Response:** The spatial patterns of these three pollution cases are different, therefore we respectively choose the corresponding sites to reflect their unique distribution characteristics. For example, the pollution during Case-2 is severe in the south and

relative lighter in the north, so the southern sites (such as Liaocheng and Xingtai) and northern sites (such as Chengde and Beijing) are presented for comparison; while for Case3, since the heavy pollution is mainly concentrated in the front of the mountains, the sites near the mountains (e.g. Shijiazhuang and Baoding) and far away from the mountains (e.g. Binzhou and Dezhou) are selected to highlight this feature. In the same way, sites such as Handan and Shijiazhuang and sites such as Cangzhou Binzhou are chosen in Case-1 to reflect the pollution expansion from the piedmont and the southwestern part to the middle and eastern area.

*[R1.12] Figure 3. Please state the figure represents observation data in the caption.*

**Response:** We have rewritten the figure caption in the revised manuscript in Lines 273-275, and indicated that the $PM_{2.5}$ concentration fields are derived from spatial interpolation of observation data.

*[R1.13] Figure 4. The figure looks very unclear. I think it should be re-plotted with making the wind vector line thinner.*

**Response:** Accepted. The figure has been re-plotted and the wind vectors have been set thinner in the revised manuscript.

*[R1.14] Table 1. Give the units of those presented variables.*

**Response:** The units of the potential temperature (K) and wind speed (m s$^{-1}$) have been added in Table 1 in the revised manuscript.

*[R1.15] Figure 6. Again, it seems that there is an intensive GPS sounding observation during October 7–12, 2014, but the author did not mention such observation in section*

**Response:** During October 7–12, 2014, the intensive GPS sounding observation is absent, and thus the routine sounding data are collected. This statement has been added in Section 2.1 in Lines 177-179 and in the figure caption in Lines 344-345.

*[R1.16] Line 314. Should be "5 &8".*
*Line 315 and Figure 7. Why do the authors present the results at 1500 LT and 2200 LT in this front category, but illustrate them at 1400 LT and 2300 LT for other categories? It is better to keep consistent.*
*Line 322, "Figs. 5 &8" is different from the description of the figure caption, please check and keep consistent.*
*Line 324. No Fig.7c-d, please add them in figure 7.*

**Response:** According to the suggestions of Reviewer #2, the paragraph "Frontal category" and the corresponding Fig. 7 in the original manuscript have been removed, because the "frontal category" is discussed by a previous study (Jin et al., 2021) rather

than the present work. We declare this research background and review the characteristics of the frontal category pollution in the Introduction in Lines 112-118 of the revised manuscript.

*[R1.16] Line 325-326. It is difficult to see the change of the PBL height; I suggest the authors add the diagnosed PBL height in Figure 7 and other figures (e.g., Fig. 8, 9, 10, 11) to better support their explanations.*

**Response:** Accepted. We have superimposed the PBL heights on the vertical cross-sections of potential temperature and wind divergence, i.e., new Figs. 7-11 and updated the figure captions in the revised manuscript. The improved figures better illustrate the evolution of the PBL thermal and dynamic structures during these pollution cases.

*[R1.17] Line 420. Should be PBL height in Jinan had increased to 1100 m.*

**Response:** Corrected as suggested.

**Response to Reviewer #2:**

We sincerely thank Reviewer #2 for the helpful comments. According to these professional suggestions, we have improved the manuscript structure and deepened the analysis of the results. The response to each comment is listed below. The original comments are in *blue and italic*, our replies are in normal font. Bracketed numbers are used for referee comments (e.g., *[R2.1]*).

*Note: the reviewer had only access to part 2 of the study. Publication of part 1 is needed before publication of part 2.*

**Response:** Recently, our two manuscripts are no longer submitted in the form of companion papers, but treated as separate independent papers, due to some technical problems encountered during the submission process. We have modified the titles (i.e. removed the label of part "1." and "2.") and rephrased related paragraphs to ensure that they are separately readable and understandable.

GENERAL COMMENTS

*In the manuscript by Jin et al., three representative cases of pollution in the North China Plain are selected and discussed. The corresponding meteorological conditions simulated by the Weather Research and Forecasting (WRF) model are presented. In a first part, these simulations are validated by comparison with the observed spatial distribution of near-surface potential temperature and wind velocity, and with the vertical profiles of wind speed and potential temperature from soundings. In the second part, the three-dimensional structure of the meteorological systems is analysed for each of the three cases with reference to the dynamics of the pollution dispersion.*

*The manuscript presents the extensive work done by the authors, which may be of interest to the scientific community, but it is not always effective in clearly explaining the relationship between the meteorological conditions and the pollution dispersion (e.g., cases 1 and 2). Additionally, the reading is further complicated by to the very poor English. The structure of the manuscript could be also improved. Hence, the manuscript can be accepted only after major changes.*

**Response:** We faithfully accept the reviewer's criticism and appreciate these valuable suggestions. We have paid special attention to these three points: (1) The influence of meteorological conditions on pollution evolution is analyzed more deeply and comprehensively to elucidate their relationship; (2) The inappropriate/vague expressions are checked and revised throughout the manuscript to ensure it is more readable and understandable; (3) The manuscript structure is improved by moving the original Fig.12 and related descriptions into the Introduction and deleting Fig. 7 and the paragraph "Frontal category". The numbering of all figures is changed accordingly. We

are sorry that there appear so many modifications to the original manuscript, but the revised manuscript does become clearer and better structured.

SPECIFIC COMMENTS

*[R2.1] The dynamics of the pollution diffusion are only addressed from a meteorological perspective, in terms of transport and horizontal/vertical dispersion processes. Emissions (e.g., their spatial distribution and their temporal variations) are not even mentioned. The authors should state why emissions are of secondary importance compared to the meteorological configuration they deepen in their manuscript, and why chemical transport models (e.g., WRF-Chem) are not needed/advised for the interpretation of these results.*

**Response:** We thank the reviewer for these critical comments. Indeed, emissions, as the fundamental cause of air pollution, should be mentioned. In the revised manuscript, we add a brief introduction of the emission patterns and their contribution to $PM_{2.5}$ pollution in this region, so that readers can get a more complete view of this pollution problem. On this basis, the significance of meteorological conditions is explained and emphasized. The revised part in Lines 90-106 is as follows. And a spatial distribution figure of the $PM_{2.5}$ emission is added in the supplement material, also shown below.

"The North China Plain (NCP) is one of the most polluted areas in the world. The dense population and developed industries produce intensive emissions in this region, with most sources located in the plain area and less in the northern and western mountains (their spatial distribution is presented in the supplement material). High-intensity primary emissions are the fundamental cause of air pollution, which directly releases pollutants into the atmosphere and provides precursors for secondary aerosol formation (Lyu et al., 2016; Zhao et al., 2019). In order to improve the air quality, a series of stringent emission reduction policies are implemented from 2013, which make the annual mean $PM_{2.5}$ concentrations decrease by 32% in 2017 (Zhang et al, 2019). However, the severe polluted days still occur frequently, especially in winter (Zhang et al., 2018). During these pollution episodes, adverse meteorological conditions are the dominant factors causing high pollution levels and various spatial patterns, as there are no significant changes in emissions in a short period (e.g., weeks). Extensive studies have been conducted to investigate the meteorological causes of regional pollution in the NCP, such as the local meteorological factors and large-scale synoptic process (Ye et al., 2016; Ren et al., 2019; Li et al., 2020). Nevertheless, the knowledge about the PBL spatial structures under the impact of the mesoscale AIBs is still insufficient, and the role of the special PBL structures plays in the air pollution evolution at a regional scale is even unclear (Bluestein, 2008; McNider and Pour-Biazar, 2020)."

[Figure]

Figure S1. Spatial distribution of monthly mean PM2.5 emission intensity in the North China Plain during wintertime of 2016 (The data comes from the website http://meicmodel.org).

Regarding the question on using only the meteorological model, Reviewer#1 also mentioned it. We simply copy the response to Reviewer#1 [R1.1] below. The relevant discussion is added in Lines 612-630 of the revised manuscript.

"The present study focuses on the mesoscale PBL structures under pollution conditions, and we try to combine the strength/advantage of both observation facts and numerical simulation capacity to investigate this issue. The densely distributed network of monitoring stations provides reliable PM2.5 pollution facts. WRF model provides boundary layer meteorology information. A chemical transport model (e.g., WRF-Chem or WRF-CMAQ) can of course simulate the whole process from meteorology to pollutants transport/diffusion and chemical transformation. But the uncertainties caused by emission inventory, chemical mechanism, and meteorology conditions may complicate the simulation results together (e.g., Travis et al., 2016; Bouarar et al., 2019; Wang et al., 2021). Our current work isolates the boundary layer meteorology simulation from other factors, and evaluates the model intensively by observation data. Therefore, this study relies more on observations, but utilizes ultimately the capacity of the meteorological model to reveal the three-dimensional structure of PBL when pollution occurs."

*[R2.2] While for case 3 the results of the meteorological simulations and the pollution dispersion dynamics are clearly explained, this is not the case, in my opinion, for cases 1 and 2. I think that the authors should better relate their numerical results with the pollution dispersion dynamics, i.e. by discussing the relation between Figs. 8-9 with Fig. 3 in the corresponding cases.*

**Response:** We have added more descriptions and analyses of Figs. 8-9 and Fig.4 (original Fig. 3) to comprehensively relate the simulated boundary layer structure with pollution dispersion dynamics in Lines 425-429, 434-435, 437-438, 440-441, 454-457, and 464-468 of the revised manuscript. In addition, the vertical cross-sections of potential temperature for Case-1 and Case-2 are presented in the supplementary materials, to illustrate that they are mainly driven by dynamic flows rather than thermal structure, according to the suggestion of Reviewer #1. After these revisions, the manuscript provides a clearer explanation of the influence of meteorological conditions on pollution evolution during these representative cases.

The added analysis in the revised manuscript and the supplementary vertical cross-sections of potential temperature (Figs. S2-S3) are as follows.

"This pollution category, mainly involving two modes of west-southwest wind shear and south-north wind shear, is driven by dynamic flows. Therefore, for the corresponding Case-1 and Case-2, the wind divergence sections are analyzed in detail in the following (Figs. 8-9). The potential temperature sections are presented in the supplementary material (Figs. S2-3), which illustrates that there is no significant thermal discontinuity.

Figure 8 displays the PBL dynamic structure of Case-1. During the pollution formation-maintenance stage, with the establishment of a low-pressure trough, westerly winds shifted to southwesterly winds at the trough axis and thus formed a convergence belt at the surface with a divergence of $-2\sim-4\times10^{-6}$ s$^{-1}$ (Fig. 8a, i). As a consequence, a mass of pollutants were transported here and further accumulated to form a pollution zone (refer to Fig. 4a, i). This trough-convergence belt continued to move to the southeast, and evolved into a cyclonic-convergence center at the end of the maintenance phase (Fig. 8a, i-iv). During this process, its affected area was expanded, so that the large range of NCP was filled with pollutants (refer to Fig. 4a, ii-iv). In the process of pollution diffusion, with the advent of a northeast high-pressure system, divergent wind fields occurred correspondently (Fig. 8a, v), which made this part of the pollutants cleaned quickly (refer to Fig. 4a, v). As for the south-north wind shear mode, the surface divergence fields displayed a "lying Y shaped" convergence zone with the opening to the west during the pollution formation-maintenance stage of Case-2 (Fig. 9a, i-iv), which was caused by the meeting of the southerly winds and the northerly winds and then turning to the easterly winds. This convergence mode made the distribution of pollutants in a pattern of much higher concentration in the south and lower in the north, with a clear edge between these two air masses (refer to Fig. 4b, i-iv). Although the southerly winds in the southern NCP kept the pollutants transported northward, they never reached the northernmost part due to the opposite airflow there."

[Figure]

Figure S2. (a) Surface spatial distributions and (b) vertical cross-sections of the simulated potential temperature at the pollution stages of (i) formation, (ii-iv) maintenance, and (v) diffusion during representative Case-1 under west-southwest wind shear mode. The black lines in (a) indicate the section lines in (b). The purple dashed lines in (b) indicate the PBL heights.

[Figure]

Figure S3. Same as Fig. S2, but for representative Case-2 under south-north wind shear mode.

*[R2.3] The structure of the paper could be improved. Figure 12 is very explanatory and, in my opinion, should be shown at the beginning of the manuscript, in order to introduce the cases. However, it should be clearly stated from the beginning that the "frontal category" is not addressed in the paper, as it has been already discussed by Jin et al. 2021 (otherwise the reader will be convinced that the three cases discussed in the manuscript are the three ones shown in the figure). Hence, the paragraph "Frontal category" at page 13 and the corresponding Fig. 7, referring to a case that is not*

*properly introduced (2 December 2017) should be removed.*

**Response:** We thank the reviewer very much for this insightful comment. The structure of the manuscript is modified according to these suggestions. The original Figure 12 is moved to the Introduction part as new Fig. 1 in the revised manuscript and the corresponding description is added in Lines 111-129 and Lines 153-165. The research status of the "frontal category" (i.e., it has been already discussed by Jin et al. 2021, rather than analyzed in this study) is declared at the beginning of the paper in Lines 112-118. The paragraph "Frontal category" and the corresponding Fig. 7 in the original manuscript have been removed. The improved structure more clearly presents the relationship between the three pollution categories reviewed and the representative cases discussed in this study.

The added description in the revised manuscript is as follows.

"Figure 1 shows the schematic diagram of three pollution categories corresponding to various AIBs. The frontal category represents about 41 % of all 98 pollution episodes, and its PBL spatial structure has been revealed in a previous case study (Jin et al., 2021). It is characterized by an isolated cold air mass, which is laterally confined by mountains and warm front AIB, and vertically covered by a warm dome (Fig. 1a). The strong elevated inversion depresses the PBL height abruptly to 200~300 m within the cold area in contrast to 600~800 m outside the zone, constituting adverse dispersion conditions and resulting in the most serious $PM_{2.5}$ pollution. The wind shear category is associated with airflow convergence AIB (Fig. 1b), which is dominated by dynamic effect and causes lighter $PM_{2.5}$ pollution. West-southwest wind shear and south-north wind shear are the two main modes. The third category occurs when the airflow cannot cross the topographic obstruction and form the cold air damming AIB. A cold and heavy pollution belt develops at the foot of the windward mountains (Fig. 1c), under the synergistic effect of dynamical obstruction and thermal stratification. Although previous studies have classified the air pollution and revealed the spatial characteristics of the first category, the three-dimensional PBL structures that interacted with AIBs under the other two categories are not yet clarified, which is responsible for 43% of pollution episodes in the NCP. In order to fulfill this knowledge gap, the present study deeply analyzes representative cases of wind shear category and topographic obstruction category (Detailed analyses in Sect 3.3), and finally provides a complete conceptual model of the PBL spatial structure in the NCP under various pollution categories and corresponding AIBs (Fig.1)."

TECHNICAL REMARKS

*[R2.4] Is "Atmospheric Internal Boundaries" common expression? I have found very few papers referring to AIB, with part of them using it with reference to the tropopause. Moreover, case 3 is essentially due to orographic obstruction, therefore I wonder if this expression should be used at all;*

**Response:** "Atmospheric Internal Boundaries" refer to the mesoscale meteorological discontinuities in the atmosphere, usually associated with temperature contrast and/or wind shift. It is firstly proposed in the research of convection triggering, and also be called "surface boundaries" (Sanders and Doswell, 1995; Hane et al., 2002; Bluestein, 2008). Their influence on the initiation of convective storms has been emphasized in these previous studies. As internal lateral boundaries within the low-level atmosphere, they lead to the discontinuity of the thermal and dynamic structures of the boundary layer, and thus play important roles in shaping the air pollution at the regional scale, as shown in this and our previous works (Jin et al., 2021; 2022).

Case 3 is indeed caused by orographic obstruction of air flow. But the horizontal structure of the lower atmosphere close to the mountains is apparent, where the localized air mass displays a distinct contrast of temperature and wind speed to its outside. This phenomenon is recognized as cold air damming (Bell and Bosart, 1988; Bailey et al., 2003; Rackley and Knox, 2016). Here we just name this kind of AIB accordingly.

The definition of the "Atmospheric Internal Boundaries" and the explanation of cold air damming AIB in Case 3 were in Lines 73-74 and Line 525-530 of the original manuscript. A more complete introduction to this concept is added in the revised manuscript in Lines 71-79, as follows.

"As the intermediate scale, mesoscale systems interact with PBL in more direct and complex ways, since they occur in the lower troposphere with vertical extension comparable with the PBL depth and horizontal scale close to the regional range. Discontinuity of meteorological properties inside and outside these systems presents as atmospheric internal boundaries (AIBs) in the lateral direction, usually manifested as temperature contrast and/or wind shift. Previous studies have emphasized their influence on the initiation of convective storms (Sanders and Doswell, 1995; Hane et al., 2002; Bluestein, 2008). On the other side, as internal lateral boundaries within the low-level atmosphere, the AIBs can lead to the abrupt change of the PBL spatial structure, which is of particular importance to the evolution of regional pollution."

*[R2.5] Large part of the abstract (e.g., classification, percentage of occurrence, etc.) describes the work done in part 1 paper (as clarified in the Introduction), therefore the abstract should be rewritten in a more specific way for the present manuscript. It should rather focus on the results of the validation and use of numerical weather simulations to explain the pollution dispersion. The first category should not even mentioned, as this was already studied in a previous publication. Mention of the sub-categories in the abstract is premature (l. 10-14 are obscure to the reader). Also, no classification (l. 2) is made in the present manuscript, but rather some representative cases are chosen and discussed;*

**Response:** We appreciate this constructive comment. The abstract has been rewritten in the revised manuscript. The original parts describing pollution classification and its occurrence frequency are removed, and the frontal category and subtypes are not mentioned again. The revised abstract focuses on describing the mesoscale

meteorological modelling and its performance, and specifically explaining the relationship between PBL structure and pollution evolution during representative cases of wind shear category and topographic obstruction category.

The revised abstract is as follows.

"This study reveals mesoscale planetary boundary layer (PBL) structures under various pollution categories during autumn and winter in the North China Plain. The role of the atmospheric internal boundaries (AIBs, referring to the discontinuity of meteorological conditions in the lateral direction) in regulating PBL structure and shaping the $PM_{2.5}$ pollution patterns is emphasized. The Weather Research and Forecast model is used to display the three-dimensional meteorological fields, and its performance is evaluated by surface observations and intensive soundings. The evaluation demonstrates that the model reasonably captures the mesoscale processes and the corresponding PBL structures. Based on the reliable simulations, three typical pollution cases are analyzed. Case-1 and Case-2 represent the two main modes of the wind shear category pollution, which is featured with airflow convergence line/zone as AIB and thus is dominated by dynamic effect. Case-1 presents the west-southwest wind shear mode associated with a trough convergence belt. The convergent airflow layer is comparable to the vertical scale of the PBL, allowing $PM_{2.5}$ accumulation to form a high pollution area. Case-2 exhibits another mode with south-north wind shear. A "lying Y-shaped" convergence zone is formed with a thickness of about 3000m, extending beyond the PBL. It defines a clear edge between the southern polluted airmass and the clean air in the north. Case-3 represents the topographic obstruction category, which is characterized by a cold-air damming AIB in front of the mountains. The PBL at the foothills is thermally stable and dynamically stagnant due to the capping inversion and the convergent winds. It is in sharp contrast to the well-mixed/ventilated PBL in the southern plain, especially in the afternoon. At night, this meteorological discontinuity becomes less pronounced. The diurnal variation of the PBL thermal-dynamic structure causes the pollutants to concentrate at the foot of the mountains during the daytime and locally accumulate throughout the entire plain in the evening. These results provide a more complete mesoscale view of the PBL structure and highlight its spatial heterogeneity, which promotes the understanding of air pollution at the regional scale."

*[R2.6] please, revise use of "the/a" articles, which are missing in many sentences, e.g. at l. 1, 7, 19, 117, 277, 351;*

**Response:** Many thanks to the reviewer for the detailed remarks. We have revised all the above-mentioned sentences and carefully checked throughout the manuscript to the best of our ability.

*[R2.7] as well "as": l. 7, 496;*

**Response:** "as well" has been corrected to "as well as" in the revised manuscript.

*[R2.8] l. 29: "is" --> "plays";*

**Response:** Corrected as suggested.

**[R2.9]** *l. 31-32: what "property"? What "variation"?*

**Response:** "Property" refers to local features such as turbulence intensity, and "variation" refers to horizontal discontinuities in wind, temperature, humidity and etc. A more complete statement in Lines 49-53 of the revised manuscript is as follows.

"The PBL structure has been recognized to be strongly dependent on three categories of factors: (i) the single-column vertical property (such as turbulence intensity) forced by the local surface's energy balance; (ii) the lateral-section horizontal variation of wind, temperature and humidity regulated by the mesoscale meteorological process and (iii) the three-dimensional spatial evolution controlled by the large-scale synoptic system."

**[R2.10]** *l. 51: Do you mean "At the intermediate scale"?*

**Response:** Yes, "At the intermediate" has been corrected to "At the intermediate scale" in Line 71 of the revised manuscript.

**[R2.11]** *l. 67, "is still insufficient": any bibliographic reference to support this sentence?*

**Response:** References are added to support the sentence in Lines 101-106 of the revised manuscript as follows.

"Extensive studies have been conducted to investigate the meteorological causes of regional pollution in the NCP, such as the local meteorological factors and large-scale synoptic process (Ye et al., 2016; Ren et al., 2019; Li et al., 2020). Nevertheless, the knowledge about the PBL spatial structures under the impact of the mesoscale AIBs is still insufficient, and the role of the special PBL structures plays in the air pollution evolution at a regional scale is even unclear (Bluestein, 2008; McNider and Pour-Biazar, 2020)"

**[R2.12]** *l. 79: missing conjunction?*

**Response:** This sentence has been rewritten as "Although previous studies have classified the air pollution and revealed the spatial characteristics of the first category, the three-dimensional PBL structures that interacted with AIBs under the other two categories are not yet clarified." in Lines 123-126 of the revised manuscript.

**[R2.13]** *l. 86: "associate" --> "associated";*

**Response:** Corrected as suggested.

**[R2.14]** *l. 88: "the" --> "an";*

**Response:** Corrected as suggested.

*[R2.15] l. 106: "was" --> "were";*

**Response:** Corrected as suggested.

*[R2.16] l. 114: "the three-point moving average method" --> "a three-point moving average", unless a more specific technique is meant here (reference needed in that case);*

**Response:** Thanks for the reviewer's detailed comment. "the three-point moving average method" has been corrected to "a three-point moving average method" in Line 187 of the revised manuscript.

*[R2.17] l. 126, 341: "pentagram" --> "star"; "pentacles" --> "stars";*

**Response:** Corrected as suggested.

*[R2.18] l. 137, "three categories/six types": it is difficult to understand the relationship between the "categories" and the "types";*

**Response:** According to comment *[R2.36]*, "type" has been no longer mentioned in the revised manuscript, and only three major categories of pollution are discussed.

*[R2.19] l. 141-142: is the frontal case was already studies in a previous paper, there is no need to recapitulate it here;*

**Response:** Accepted. The description of the frontal case in Lines 141-142 of the original manuscript has been removed.

*[R2.20] l. 142-145: too much detail relative to part 1. The reader should be able to understand this manuscript independently from the first part;*

**Response:** Thanks for the reviewer's constructive suggestion. This paragraph has been rephrased as follows to make the manuscript independently understandable (in Lines 212-227 of the revised manuscript).

"As mentioned above, PM$_{2.5}$ pollution episodes in the NCP are identified in the frontal category, wind shear category, and topographic obstruction category, according to their association with the mesoscale AIBs (Jin et al. 2022 submitted). The present study tries to reveal the PBL structures modified by the AIBs under various pollution categories. Among them, the first category has been investigated previously (Jin et al., 2021). We focus on the representative cases under the other two categories in this paper. For the wind shear category, there are two main shear modes: west-southwest wind shear and south-north wind shear. Therefore a total of three typical cases are selected

to respectively represent these two pollution categories, i.e., Case-1 for west-southwest wind shear mode: during January 17–21, 2018; Case-2 for south-north wind shear mode: during January 7–11, 2016; and Case-3 for topographic obstruction category: during October 7–12, 2014."

*[R2.21] l. 156: the two peaks are not clear in all sites. Also, Fig. 2a does not show the formation stage (increasing concentrations) for most sites;*

**Response:** The locations of the two concentration peaks have been pointed out in Lines 237-240 of the revised manuscript. Figure. 2 has been redrawn as new Fig. 3 to show the pollution formation stage of Case-1. For your convenience, the revised sentence and the modified Fig. 3 are presented as follows.

"As shown in Fig. 3a, Case-1 was characterized by two main concentration peaks (300 µg m$^{-3}$ at Handan vs 500 µg m$^{-3}$ at Cangzhou) in the formation-maintenance stage (January 17–20, 2018), with the latter being higher than the former."

[Figure]

Figure 3. Temporal evolution of PM$_{2.5}$ concentrations during Case1–3, respectively represent (a) west-southwest wind shear mode (January 17–21, 2018), (b) south-north wind shear mode (January 7–11, 2016), and (c) topographic obstruction category (October 7–12, 2014). The locations of these PM$_{2.5}$ stations are marked in Fig. 2a.

*[R2.22] l. 170-173: for case 1, the build-up seems to start also at the south-west side, not only along the mountains. For the same reason, it is difficult to state that the "pollution center has been transferred eastward";*

**Response:** The vague description has been rewritten in Lines 251-256 of the revised manuscript, as follows.

"In the formation stage, the polluted air mass of Case-1 and Case-3 built up along the mountains from the southwest of the NCP, with the latter being more concentrated and the former spreading southwestward (Fig. 4a-i, c-i). While the pollution in Case-2 first developed from the south (Fig. 4b-i). During the pollution maintenance process, Case-1 was featured with widespread $PM_{2.5}$ flooding the NCP, making the eastern region gradually covered by heavy pollution (Fig. 4a, ii-iv)"

*[R2.23] l. 205: "... southern edges"?*

**Response:** The confusing sentence has been rephrased as "As a result, the polluted air mass was prevented from advancing northward to the mountains, causing a strong contrast in pollution concentration between the northern and southern parts of the domain" in the revised manuscript in Lines 297-299.

*[R2.24] l. 206 and 256: is "high-pressure invasion" a Chinese idiom?*

**Response:** We have corrected this inappropriate expression as "high-pressure system" in Line 300 of the revised manuscript.

*[R2.25] l. 215-218: clearly state that these are observations and that they are spatially interpolated based on Jin et al. 2021;*

**Response:** We have rewritten the figure caption in the revised manuscript in Lines 273-275, and indicated that the $PM_{2.5}$ concentration fields are derived from spatial interpolation of observation data.

*[R2.26] l. 189: rephrase this sentence, it is unclear;*

**Response:** This sentence has been rephrased in Lines 282-283 of the revised manuscript as follows.

"Case-1 and Case-2 are the two main modes of wind shear category, for which dynamic AIB plays a dominant role."

*[R2.27] Figs. 4, 5, 8, 9, and 10 are too small. Consider rotating them by 90° and displaying them at full page;*

**Response:** Accepted. These figures are displayed on the full page in the revised manuscript to present a more clear view.

*[R2.28]* *l. 240: the large correlation coefficient of the potential temperature may be simply due to the day/night cycle, which is common in both the model and the observations, thus it is not representative of the model performances;*

**Response:** Thanks for the reviewer's professional comment. The correlation coefficients of the daily averages of potential temperature and wind speed are added in Lines 326-328 of the revised manuscript to exclude the influence of the diurnal cycle. A corresponding table is included in the supplementary material to show the detailed statistics of the daily averages. Related descriptions and Table S1 are shown below.

"In order to exclude the influence of the diurnal cycle on the correlation, the daily averages are also calculated and the obtained correlation coefficients are as high as 0.65~1 and 0.62~1 (p<0.01) for potential temperature and wind speed, respectively (Table S1)."

Table S1. Statistics of model performance for the daily average near-surface potential temperature and 10 m wind speed for selected 13 cities during the representative cases.

| | Case-1 | | | | Case-2 | | | | Case-3 | | | |
|---|---|---|---|---|---|---|---|---|---|---|---|---|
| | PT (K) | | WS (m s$^{-1}$) | | PT (K) | | WS (m s$^{-1}$) | | PT (K) | | WS (m s$^{-1}$) | |
| | R | RMSE | R | RMSE | R | RMSE | R | RMSE | R | RMSE | R | RMSE |
| Beijing | 0.99 | 0.94 | 0.91 | 0.65 | 0.89 | 2.99 | 0.72 | 1.67 | 0.74 | 1.84 | 0.94 | 1.20 |
| Tianjin | 0.99 | 1.14 | 0.99 | 1.03 | 0.96 | 2.46 | 0.67 | 1.48 | 0.90 | 1.71 | 0.62 | 1.90 |
| Shijiazhuang | 0.91 | 1.51 | 0.73 | 1.83 | 0.99 | 3.47 | 0.95 | 1.72 | 0.96 | 0.63 | 0.91 | 1.67 |
| Baoding | 0.79 | 0.73 | 0.84 | 0.82 | 0.98 | 2.55 | 0.99 | 1.03 | 0.93 | 1.34 | 0.83 | 1.59 |
| Handan | 0.98 | 1.00 | 1.00 | 0.19 | 0.90 | 2.33 | 0.96 | 1.14 | 0.99 | 0.69 | 0.78 | 1.64 |
| Tangshan | 0.86 | 1.61 | 0.94 | 0.77 | 1.00 | 3.14 | 0.98 | 0.53 | 0.71 | 2.45 | 0.89 | 2.00 |
| Cangzhou | 0.83 | 1.94 | 0.98 | 0.61 | 0.74 | 1.34 | 0.97 | 0.85 | 0.98 | 1.57 | 0.94 | 1.15 |
| Dezhou | 0.80 | 2.49 | 0.69 | 1.28 | 0.72 | 3.27 | 1.00 | 1.71 | 0.98 | 1.20 | 0.87 | 2.77 |
| Jinan | 0.79 | 1.61 | 0.80 | 1.46 | 0.99 | 3.71 | 0.99 | 1.57 | 0.94 | 1.17 | 0.66 | 2.38 |
| Weifang | 0.66 | 0.93 | 0.66 | 1.28 | 0.65 | 3.01 | 0.99 | 2.19 | 0.83 | 1.37 | 0.99 | 1.12 |
| Binzhou | 0.69 | 0.88 | 0.69 | 1.51 | 0.71 | 2.57 | 1.00 | 1.59 | 0.68 | 1.21 | 0.95 | 1.18 |
| Chengde | 0.72 | 3.66 | 0.76 | 1.55 | 0.70 | 5.14 | 0.97 | 0.80 | 0.81 | 2.87 | 0.96 | 1.29 |
| Zhangjiakou | 0.86 | 4.15 | 0.87 | 0.90 | 0.70 | 4.46 | 0.74 | 1.87 | 1.00 | 4.66 | 0.86 | 2.09 |
| **Average** | **0.84** | **1.74** | **0.84** | **1.07** | **0.83** | **3.11** | **0.92** | **1.40** | **0.88** | **1.75** | **0.86** | **1.69** |

Case-1: west-southwest wind shear mode (January 17–21, 2018); Case-2: south-north wind shear mode (January 7–11, 2016); Case3: topographic obstruction category (October 7–12, 2014).

*[R2.29]* *l. 250-251: rephrase;*

**Response:** This sentence has been rephrased in Lines 347-348 of the revised manuscript as follows.

"Compared with Fig. 5, the simulated surface meteorological fields during the three cases are displayed in Fig. 6."

*[R2.30] l. 260-262: what "area"? Also, the main clause is missing;*

**Response:** "area" is the simulated cold zone. The unclear statement has been rewritten as "In the simulation field, the cold zone is shorter at its south end on the afternoon of October 08, 2014, and there is an overestimate of the potential temperature in the northwest mountains and the Bohai Sea at night." in the revised manuscript in Lines 357-359.

*[R2.31] l. 264-265: "can be" --> "is";*

**Response:** Corrected as suggested.

*[R2.32] l. 280: "being" --> "playing";*

**Response:** Corrected as suggested.

*[R2.33] l. 308: "critical to" --> "critical for";*

**Response:** Corrected as suggested.

*[R2.34] l. 309-310: instead of listing all cases, wouldn't it be simpler to just say "For all cases"?*

**Response:** The redundant statement has been removed in the revised manuscript.

*[R2.35] l. 324: I cannot see any Figs. 7c-d;*

**Response:** As suggested, the original Figure 7 has been deleted in the revised manuscript.

*[R2.36] l. 332-336: if subtypes are not introduced, then rephrase without mentioning them;*

**Response:** Accepted, subtypes are no longer mentioned in the revised manuscript.

*[R2.37] l. 365: "left" --> "west";*

**Response:** Corrected as suggested.

*[R2.38] l. 374 and 478: please, explain what you mean by "sub-synoptic scale characteristics/features";*

**Response:** The "sub-synoptic scale characteristic/feature" refers to the larger-end of the mesoscale meteorology in our manuscript. Actually, there are overlays between

mesoscale and synoptic scale processes. The dynamic property of Case-3 involves a saddle-shaped pressure field with a horizontal scale of hundreds to one thousand kilometers, and the vertical scale extends beyond the boundary layer depth to a height of about 3000 m. This phenomenon seems to exceed the range of typical mesoscale, but is smaller than the typical synoptic scale, so we refer to it as a sub-synoptic feature.

**[R2.39]** *l. 394: "extracted" or "shown";*

**Response:** The redundant word "extracted" has been removed in Line 497 of the revised manuscript.

**[R2.40]** *l. 430, "more susceptible to the local property": unclear;*

**Response:** The unclear statement has been rewritten as "…the nocturnal boundary layer was stable over the whole domain and more susceptible to the local property, such as surface heterogeneity, meandering motions, and gravity waves (Mahrt, 1998)." in the revised manuscript in Lines 536-538.

**[R2.41]** *l. 442-444: grammatically inconsistent, please rephrase;*

**Response:** This sentence has been rewritten as "This study investigated the three-dimensional PBL structures modified by mesoscale AIBs under various pollution categories by using the mesoscale meteorological model WRF." in Lines 549-550.

**[R2.42]** *l. 460-470: this case is not discussed here, please remove this part;*

**Response:** Although this case is not discussed in this work, we intended to present a retrospective summary combining the previous findings. Therefore, the overview of the frontal category is retained. Moreover, a sentence is added to state this intent in Lines 562-565 of the revised manuscript, as follows.

"The results of this paper, together with a previous systematic classification study (Jin et al., 2022 submitted) and a detailed case study for frontal category (Jin et al., 2021), depict a more complete and clearer view of the PBL spatial structures during pollution episodes in the regional scale of NCP…"

**[R2.43]** *l. 502: "roughly" --> "rough".*

**Response:** Corrected as suggested.

**References**

Bailey, C. M., Hartfield, G., Lackmann, G. M., Keeter, K. and Sharp, S.: An objective climatology, classification scheme, and assessment of sensible weather impacts

for Appalachian cold-air damming. Wea. Forecasting, 18, 641–661, doi:10.1175/1520-0434(2003)018,0641:AOCCSA.2.0.CO;2,2003.

[revised manuscript text omitted]

---

## Author Response (AR2)

**Response to Reviewer #1:**

We are grateful to the reviewer for the professional comments. According to these comments, we have carefully revised the manuscript again. Additional work has been carried out to run a chemical transport model and actually show the dynamical transport processes of air pollutants. The response to each comment is listed below. The original comments are in *blue and italic*, our replies are in normal font. Bracketed numbers are used for referee comments (e.g., *[R1.1]*).

**Summary**

*My comments have been addressed partially in the revised version. The revised manuscript has improved as compared with the first submission. However, there are still issues that I consider fundamental which were not fully satisfied.*

**Response:** We appreciate the positive evaluation of our efforts, and faithfully accept the major criticism of our previous revision. We have simulated the air pollution transport processes by the WRF-Chem model to fulfill the gap between meteorology and air pollution concentration. And the manuscript is revised accordingly.

**Major comments**

*[R1.1] Although the authors give some explanations as to why they did not choose the chemical transport model (e.g., WRF-Chem) for their simulations, their WRF results do not explain the formation of pollution in cases 1 and 2 well in my view, probably because the observed PM2.5 distributions (Fig. 4) are not necessarily due to the dynamical mechanisms proposed by the authors, and therefore, the use of chemical transport models could be a good support to explain these dynamical causes. For example, in Figs. 9a i-iv of case 2, their spatial fields of wind and divergence are similar, but their PM2.5 distributions (Figs. 4b i-iv) are clearly different. A similar pattern is also found in Figure 8a i-iv. These may be better explained if the results of the chemical transport model are used.*

**Response:** Yes, there is indeed a gap between the modeled meteorological fields and the observed $PM_{2.5}$ fields shown in the manuscript. Most importantly, the meteorological fields are not correspondent to the $PM_{2.5}$ pollution distributions apparently. Therefore, we have accepted this criticism and employed the WRF-Chem model to analyze the dynamical causes of the pollution formation in Case-1 and Case-2.

We add a figure in the new version of the manuscript to show the simulation results, as new Fig. 10. It shows that wind convergence leads to the pollutants accumulation, e.g., Fig.10a vs b, and Fig.10e vs f, for Case-1 and Case-2. The concentration increment fields are presented in Fig. 10c&g, and the patterns of $PM_{2.5}$ horizontal advection

integrated over this period are shown in Fig. 10d&h. Their spatial distribution manifests a good correspondence. Quantitatively, the dynamical advection process contributes 27%-80% to the concentration increases during the pollution development period of these two cases. These results demonstrate clearly that airflow convergence plays a dominant role in the regional air pollution formation of the wind shear category.

New Fig. 10 and the detailed analysis added in Lines 412-444 of the revised manuscript are presented below.

[Figure]

Figure 10. Simulated near-surface PM2.5 concentrations (a-b, e-f) at two instants during the pollution formation-maintenance stage and their difference (c, g), as well as the temporal integration of the PM2.5 horizontal advection term over this stage (d, h) for Case-1 (upper) and Case-2 (lower) respectively.

"To provide explicit support to the above explanation between the dynamical convergence feature and the pollution development, we adopt a chemical transport model (WRF-Chem) to simulate the PM2.5 pollution process and directly quantify the advection term in the PM2.5 concentration prognostic equation, i.e.:

$$\frac{\partial c}{\partial t} = -\nabla \cdot \left(\vec{U}c\right)_{adv} + \nabla \cdot (K_e \nabla c)_{diff} + E_{emiss} + S_{sink} + R_{chem}, \tag{1}$$

where $c$ is PM2.5 concentration, $\vec{U}$ is the wind vector, $K_e$ is the turbulent diffusion coefficient. The first term on the right side of the equation represents the advection process both horizontally and vertically. The second term is turbulent diffusion, and the last three terms represent emissions, deposition and chemical reactions, respectively. The present study pays attention to the horizontal advection, which is considered of most important effect on the pollution development for the wind shear category. Details of the model configuration and validation are described in the supplementary material (Text S1, Fig. S4, and Table S2). The simulations of Case-1 and Case-2 well reproduce

the PM$_{2.5}$ pollution concentration patterns and their evolution. Their pollution formation and maintenance stages are discussed here. For Case-1, the simulated near-surface PM$_{2.5}$ fields at 14:00 of both January 18 and January 19, 2018, as well as their difference are displayed in Fig. 10a-c, indicating that the air pollution aggravates and spreads eastward. The temporal integration of the PM$_{2.5}$ horizontal advection term over this period (Fig. 10d) agrees well with the concentration increment pattern in Fig. 10c, demonstrating the crucial role of the dynamical convergence in the development of PM$_{2.5}$ pollution. The contribution of the horizontal advection term on the total increment of PM$_{2.5}$ concentration during this period over most of this region is very high, e.g., at Handan, Shijiazhuang, Baoding, and Tianjin, the contribution ranges 40%-85%. For Case-2, heavy pollution is transferred to the north and east from January 08 to January 09, 2016 (Fig. 10e-g). Similar to Case-1, the advection term integrated over the pollution formation-maintenance period (Fig. 10h) presents good agreement with the PM$_{2.5}$ increment pattern (Fig. 10g). Quantitatively, this term contributes to total concentration accumulation as high as 27%-80% in the pollution process, especially in Beijing, Tianjin, and Baoding. This result is also consistent with those in previous works (Jiang et al., 2015; Chang et al., 2019; Jin et al., 2020). The above analysis indicates that, the airflow convergence AIB does not sharply confine the pollution air mass, but provides a circumstance or structure for pollutants transporting/accumulating along or nearby this zone. Because of the dynamical property, the concentration fields of the wind shear category pollution are more variable in space and time."

*[R1.2] Also Figures 5 and 6 in the revised version are still very unclear, making it difficult for the reader to get the appropriate information. Please re-plot these figures with the same color bar and with high resolution for better comparison.*

**Response:** Accepted. The figures have been re-plotted using the same color bar and with higher resolution in the revised manuscript.

**Minor comments**

*[R1.3] Line 56. Should be "Petäjä et al., 2016", not "Petaja et al., 2016".*

**Response:** We are sorry for this mistake. "Petaja et al., 2016" has been corrected to "Petäjä et al., 2016" in Line 38 of the revised manuscript.

*[R1.4] Line 93. Please specific the figure number in the supplement material for the emission spatial distribution.*

**Response:** Accepted. The figure number "Fig. S1" in the supplementary material for the emission spatial distribution has been specified in Line 73 of the revised manuscript.

*[R1.5] Line 435. Can you explain how the reader can see such a low-pressure trough?*

**Response:** The low-pressure trough is shown in Fig. 6a, presenting a narrow area extending from a low-pressure system with the surface isobar opening to the north. We have added the word "(refer to Fig. 6a)" in Line 367 of the revised manuscript to make the reader clearly see the low-pressure trough.

*[R1.6] Line 612. Should be "The present study focuses on the characteristic of mesoscale PBL structures"?*

**Response:** Yes. This sentence has been corrected in Line 557 of the revised manuscript.

**References**

Chang, X., Wang, S.X., Zhao, B., Xing, J., Liu, X.X., Wei, L., Song, Y., Wu, W.J., Cai, S.Y., Zheng, H.T., Ding, D. and Zheng, M.: Contributions of inter-city and regional transport to PM2.5 concentrations in the Beijing-Tianjin-Hebei region and its implications on regional joint air pollution control, Sci. Total Environ. 660, 1191-1200, doi:10.1016/j.scitotenv.2018.12.474, 2019.

Jiang, C., Wang, H., Zhao, T., Li, T., Che, H.: Modeling study of PM2.5 pollutant transport across cities in China's Jing–Jin–Ji region during a severe haze episode in December 2013. Atmos. Chem. Phys. 15, 2969–2983, doi:10.5194/acp-15-5803-2015, 2015.

Jin, X. P., Cai, X. H., Yu, M. Y., Song, Y., Wang, X. S., Kang, L. and Zhang, H. S.: Diagnostic analysis of wintertime PM2.5 pollution in the North China Plain: The impacts of regional transport and atmospheric boundary layer variation. Atmos. Environ., 224, 117346, doi: 10.1016/j.atmosenv.2020.117346, 2020.